# Application of the Multi-Scale Infrastructure for Chemistry and Aerosols version 0 (MUSICAv0) for air quality research in Africa

**Wenfu Tang[1], Louisa K. Emmons[1], Helen M. Worden[1], Rajesh Kumar[2], Cenlin He[2], Benjamin Gaubert[1], Zhonghua Zheng[3], Simone Tilmes[1], Rebecca R. Buchholz[1], Sara-Eva Martinez-Alonso[1], Claire Granier[4,5], Antonin Soulie[4], Kathryn McKain[6], Bruce C. Daube[7], Jeff Peischl[5,8], Chelsea Thompson[8], and Pieternel Levelt[1,9,10]**

[1]Atmospheric Chemistry Observations & Modeling Laboratory, National Center for Atmospheric Research, Boulder, CO, USA

[2]Research Applications Laboratory, National Center for Atmospheric Research, Boulder, CO, USA

[3]Department of Earth and Environmental Sciences, The University of Manchester, Manchester M13 9PL, United Kingdom

[4]Laboratoire d'Aérologie, CNRS, Université de Toulouse, Toulouse, France
[5]Cooperative Institute for Research in Environmental Sciences (CIRES), University of Colorado, Boulder, CO, USA
[6]Global Monitoring Laboratory (GML), National Oceanic and Atmospheric Administration, Boulder, CO, USA
[7]Department of Earth and Planetary Sciences, Harvard University, Cambridge, MA, USA
[8]NOAA Chemical Sciences Laboratory, Boulder, CO, USA
[9]Royal Netherlands Meteorological Institute (KNMI), Utrechtseweg 297, 3730 AE De Bilt, the Netherlands
[10]University of Technology Delft, Mekelweg 5, 2628 CD Delft, the Netherlands

Correspondence: Wenfu Tang (wenfut@ucar.edu)

**Abstract**
The Multi-Scale Infrastructure for Chemistry and Aerosols Version 0 (MUSICAv0) is a new community modeling infrastructure that enables the study of atmospheric composition and chemistry across all relevant scales. We develop a MUSICAv0 grid with Africa refinement (~28 km × 28 km over Africa). We evaluate the MUSICAv0 simulation for 2017 with in situ observations and compare the model results to satellite products over Africa. A simulation from the Weather Research and Forecasting model coupled with Chemistry (WRF-Chem), a regional model that is widely used in Africa studies, is also included in the analyses as a reference. Overall, the performance of MUSICAv0 is comparable to WRF-Chem. Both models underestimate carbon monoxide (CO) compared to in situ observations and satellite CO column retrievals from the Measurements of Pollution in the Troposphere (MOPITT) satellite instrument. MUSICAv0 tends to overestimate ozone ($O_3$), likely due to overestimated stratosphere-to-troposphere flux of ozone. Both models significantly underestimate fine particulate matter ($PM_{2.5}$) at two surface sites in East Africa. The MUSICAv0 simulation agrees better with aerosol optical depth (AOD) retrievals from the Moderate Resolution Imaging Spectroradiometer (MODIS) and tropospheric nitrogen dioxide

(NO$_2$) column retrievals from the Ozone Monitoring Instrument (OMI) than WRF-Chem.
MUSICAv0 has a consistently lower tropospheric formaldehyde (HCHO) column than OMI
retrievals. Based on model-satellite discrepancies between MUSICAv0 and WRF-Chem and
MOPITT CO, MODIS AOD, and OMI tropospheric NO$_2$, we find that future field campaign(s)
and more in situ observations in an East African region (30°E – 45°E, 5°S – 5°N) could
substantially improve the predictive skill of atmospheric chemistry model(s). This suggested focus
region exhibits the largest model-in situ observation discrepancies, as well as targets for high
population density, land cover variability, and anthropogenic pollution sources.

## 1. Introduction

As one of the most dramatically changing continents, Africa is experiencing myriad
environmental sustainability issues (e.g., Davidson et al., 2003; Washington et al., 2006; Ziervogel
et al., 2014; Boone et al., 2016; Swilling et al., 2016; Baudoin et al., 2017; Güneralp et al., 2017;
Nicholson 2019; Fisher et al., 2021; Langerman et al., 2023). These environmental issues are
causing vast losses in lives and in African economies, and are coupled with poverty and under-
development (Washington et al., 2006; Fisher et al., 2021). Some of these environmental
challenges are particularly severe in Africa compared to many other regions of the world (e.g.,
droughts, floods, high temperatures, land degradation, and fires; Washington et al., 2006; Nka et
al., 2015; van der Werf et al., 2017; Haile et al., 2019). However, even though Africa is the second
largest continent, in land area and population, attention and research on environmental challenges
in Africa are very limited, leading to a deficit of knowledge and solutions (e.g., De Longueville et
al., 2010). Intergovernmental Panel on Climate Change (IPCC) computes a human vulnerability
metric from existing challenges such as poverty, access to health care plus expected mortality for
climate hazards such as heat, drought, flood, fires and constraints to adaptation like funding, and
government infrastructure (Moss et al., 2001). Many regions in Africa exhibit the most extreme
values for this metric.
Degraded air quality is an example of a severe environmental challenge with growing
importance in Africa (e.g., Kinney et al., 2011; Naiker et al., 2012; Liousse et al., 2014; Thompson
et al., 2014; Amegah et al., 2017; Heft-Neal et al., 2018; Fisher et al., 2021; Okure et al., 2022;
Vohra et al., 2022). A previous study found that air pollution across Africa caused ~1.1 million
deaths in 2019 (Fisher et al., 2021). However, the study of air quality in Africa is hindered by the
scarcity of ground-based observations (e.g., Paton-Walsh et al., 2022; Kalisa et al., 2023),
modelling capability and the use of satellite observations. In this paper, we will focus on air quality
analyses over Africa with the new model Multi-Scale Infrastructure for Chemistry and Aerosols
(MUSICA; Pfister et al., 2020).
Atmospheric chemistry modeling is a useful tool to provide air quality forecasts and to
understand chemical processes. Various models have been applied to study atmospheric chemistry
and air quality in Africa such as the Weather Research and Forecasting (WRF) model coupled with
Chemistry (WRF-Chem) (e.g., Kuik et al., 2015; Kumar et al., 2022; Jenkins and Gueye, 2022),
the GEOS-Chem chemical transport model (e.g., Marais et al., 2012, 2019; Lacey et al., 2018), the
CHIMERE chemical transport model (e.g., Menut et al., 2018; Mazzeo et al., 2022), and the U.K.
Earth System Model (UKESM1) (Brown et al., 2022), and GEOS5 (Bauer et al., 2019).
MUSICA is a new state-of-the-art community modeling infrastructure that enables the
study of atmospheric composition and chemistry across all relevant scales (Pfister et al., 2020).
The newly developed MUSICA Version 0 (MUSICAv0) is a global chemistry-climate model that
allows global simulations with regional refinement down to a few kilometers spatial resolution
(Schwantes et al., 2022). The coupling with other components of the Earth system (e.g., land,
ocean, and sea ice) can also be performed at multiple scales. MUSICAv0 has various advantages
and is particularly suitable for research applications over Africa. For example, MUSICAv0 can be
used to study the interactions between atmospheric chemistry and other components of the Earth
system and climate. MUSICA also includes the whole atmosphere (from the surface to
thermosphere), and therefore can also be used to study the stratosphere and above and interactions
between the stratosphere and troposphere. This is critical because some of the environmental issues
are coupled (e.g., the ozone–climate penalty; Brown et al., 2022). In addition, as a global model,
MUSICAv0 does not require boundary conditions to study a region at high resolution. Global
impacts and interactions can be simulated in a consistent and coherent way. This feature is
important as inflow from other continents and oceans significantly impacts air quality in Africa.
MUSICAv0 has been evaluated over North America (Schwantes et al., 2022, Tang et al., 2022)
and is also being developed and tested in other regions around the globe
(https://wiki.ucar.edu/display/MUSICA/Available+Grids).
This paper serves as the basis for the future application of MUSICAv0 in Africa. In this
study, we develop a MUSICAv0 model grid with regional refinement over Africa. Because
MUSICAv0 with Africa refinement is newly developed while WRF-Chem has been previously
used for African atmospheric chemistry and air quality studies, here we include results from WRF-
Chem to assess the ability of MUSICAv0 in reproducing the regional features of atmospheric
composition as simulated by WRF-Chem. We conduct the MUSICAv0 simulation for the year
2017 to compare with a previous WRF-Chem simulation (Kumar et al., 2022). MUSICAv0 and
the WRF-Chem simulation and the observational data used in this study are described in Section
2. The MUSICAv0 model simulation results are evaluated against in situ observations and
compared with satellite retrievals in Section 3. In Section 4, we provide an example application of
MUSICAv0 over Africa – identifying key potential regions in Africa for future in situ observations
and field campaign(s).
**2. Model and data**
**2.1 MUSICAv0**

MUSICA is a newly developed framework for simulations of large-scale atmospheric
phenomena in a global modeling framework, while still resolving chemistry at emission- and
exposure-relevant scales (Pfister et al., 2020). MUSICA version 0 (MUSICAv0) is a configuration
of the Community Earth System Model (CESM). It is also known as the Community Atmospheric
Model with chemistry (CAM-chem) (Tilmes et al., 2019; Emmons et al., 2020) with regional
refinement (RR) down to a few kilometers (Lauritzen et al., 2018; Schwantes et al., 2022). CAM-
chem, and thus MUSICAv0, includes several choices of chemical mechanisms of varying
complexity. This study uses the default MOZART-TS1 chemical mechanism for gas phase
chemistry (including comprehensive tropospheric and stratospheric chemistry; Emmons et al.,
2020) and the four-mode version of the Modal Aerosol Module (MAM4; Liu et al., 2016) for the
aerosol scheme. The generation of desert dust particles in MUSICAv0 is calculated based on the
Dust Entrainment and Deposition Model (Mahowald et al., 2006; Yoshioka et al., 2017). Dust
emissions calculation is sensitive to the model surface wind speed. The dust aerosol processes in
the MUSICAv0 simulation are simulated based on the MAM4 model (Liu et al., 2016). MAM4
has 4 modes – Aitken, accumulation coarse, and primary carbon modes. Dust is mostly in the
accumulation and coarse modes. The MUSICAv0 model source code and the model
documentation           can           be           downloaded           through
https://wiki.ucar.edu/display/MUSICA/MUSICA+Home (last access: 3 April 2023).

The MUSICAv0 users have the option to create their own model grid. MUSICAv0 is
currently being developed and tested for applications over various regions globally
(https://wiki.ucar.edu/display/MUSICA/Available+Grids), including North America, India, East
Asia, South America, Australia, and Korea, among others. (e.g., Schwantes et al., 2022; Tang et
al., 2022; Jo et al., 2023). In this study, we develop a model grid for applications in Africa
(ne0np4.africa_v5.ne30x4). As shown in Figure 1a, the horizontal resolution is ~111 km × 111 km
(i.e., 1° latitude × 1° equatorial longitude) globally, and ~28 km × 28 km (i.e., 0.25° latitude ×
0.25° equatorial longitude) within the region over Africa. Our simulation uses the default option
for vertical layers (i.e., 32 layers from the surface to ~3.64 hPa).

Here we run MUSICAv0 with the model grid for Africa for the year 2017, saving 3-hourly
output. We use the Copernicus Atmosphere Monitoring Service Global Anthropogenic emissions,
(CAMS-GLOB-ANTH) version 5.1 (Soulie et al., 2023) for anthropogenic emissions and the
Quick Fire Emissions Dataset (QFED) for fire emissions (Darmenov and da Silva, 2013). CAMS-
GLOB-ANTH version 5.1 emissions can be found at https://eccad3.sedoo.fr/data (last access: 3
April     2023).     QFED     emissions     can     be     found     at
https://portal.nccs.nasa.gov/datashare/iesa/aerosol/emissions/QFED/ (last access: 3 April 2023).
CAMS-GLOB-ANT version 5.1 (Soulie et al., 2023) is one of the most widely used global
inventories for anthropogenic emissions. CAMS-GLOB-ANT version 5.1 has been implemented
in MUSICAv0, and evaluated in our previous studies (Tang et al., 2022, 2023; Jo et al., 2023).
CAMS-GLOB-ANT version 5.1 does not include information from the Dynamics-Aerosol-
Chemistry-Cloud Interactions in West Africa (DACCIWA) project, however, a future version of
CAMS-GLOB-ANT is expected to include DACCIWA for Africa. In future work on this topic,
we plan to make use of regional emissions inventories, such as the DACCIWA emission inventory.
Plume rise climatology is applied to fire emissions following Tang et al. (2022). In addition, we
also include open waste burning (https://www.acom.ucar.edu/Data/fire/; Wiedinmyer et al., 2014)
emissions in the simulation. The model has the option of a free-running atmosphere or nudging to
external meteorological reanalysis. In this simulation, only wind and temperature are nudged to
the Modern-Era Retrospective analysis for Research and Applications, Version 2 (MERRA-2;
Gelaro et al., 2017) with a relaxation time of 12 hours. MERRA-2 data can be found at
https://disc.gsfc.nasa.gov/datasets?project=MERRA-2 (last access: 3 April 2023).

We also added carbon monoxide (CO) tracers in the simulation to understand the source
and transport of air pollution. CO tracers in CAM-chem/MUSICAv0 are described in detail by
Tang et al. (2019). In this study we include tracers for 6 regions (North Africa, West Africa, East
Africa, Central Africa, Southern Africa, and the rest of the world) and 3 emission sources
separately (anthropogenic emissions, fire emissions, and open waste burning emissions). In total,
there are 18 tagged CO tracers.
**2.2 WRF-Chem**
The Weather Research and Forecasting (WRF) model coupled with Chemistry (WRF-
Chem) is a regional chemical transport model. It has been widely used for air quality studies in
Africa. In this study we use model results from a WRF-Chem simulation described by Kumar et
al. (2022). The WRF-Chem simulation has a grid spacing of 20 km, slightly higher than the
MUSICAv0 simulation, and the model domain is highlighted in Figure 1a. The simulation has 36
vertical levels from the surface to ~50 hPa. The WRF-Chem simulation uses the Model for Ozone
and Related Tracers-4 (MOZART-4) chemical mechanism (Emmons et al., 2010) for tropospheric
gas phase chemistry, and the Goddard Global Ozone Chemistry Aerosol Radiation and Transport
(GOCART) model (Chin et al., 2002) for aerosol processes. The dust aerosol processes in the
WRF-Chem simulation are simulated based on the Goddard Global Ozone Chemistry Aerosol
Radiation and Transport (GOCART) model (Chin et al., 2002). Specifically, the dust emission
scheme is following the GOCART emission treatment (Ginoux et al., 2001), which is a function
of 10-m wind speed, soil moisture, and soil erosion capability. The atmospheric processes of dust
are simulated based on the mass mixing ratio and size distribution that has been divided into 5 size
bins with effective radii of 0.73, 1.4, 2.4, 4.5 and 8.0 μm. The dust dry and wet depositions are
also treated following the GOCART scheme (Chin et al., 2002). The European Centre for Medium
Range Weather Forecasts (ECMWF) global reanalysis (ERA-Interim) fields are used for initial
and boundary meteorology conditions, while another CAM-chem simulation is used for initial and
boundary chemical conditions (Kumar et al., 2022). The WRF-Chem simulation used the global
Emission Database for Atmospheric Research developed for Hemispheric Transport of Air
Pollution (EDGAR-HTAP v2) for anthropogenic emissions and the Fire Inventory from NCAR
version 1.5 (FINNv1.5) (Wiedinmyer et al., 2011) for fire emissions. The WRF-Chem output is
saved hourly, however we only use 3-hourly output to match the MUSICAv0 simulation.
**2.3 ATom**
The Atmospheric Tomography mission (ATom; Thompson et al. 2022) was designed to
study the impact of human-produced air pollution on greenhouse gases, chemically reactive gases,
and aerosols in remote ocean air masses. ATom data (Wofsy et al., 2021) are available at
https://espoarchive.nasa.gov/archive/browse/atom (last access: 3 April 2023). During the project,
the DC-8 aircraft sampled the remote troposphere with continuous vertical profiles. There were
four seasonal deployments from the summer of 2016 through the spring of 2018. Here we compare
the MUSICAv0 simulation with observations from ATom-2 (January–February 2017) and ATom-
3 (September–October 2017). Since the ATom flight tracks were mostly outside the WRF-Chem
domain (Figure 1a), we do not compare the WRF-Chem simulation with ATom data. However,
we compare chemical species from the MUSICAv0 simulation to the 2-minute merged ATom
measurements globally to obtain a benchmark and broader understanding of MUSICAv0
performance both within and outside the refined region. The model output is saved along the ATom
aircraft flight tracks and with respect to the observational times at run time. Nitric oxide (NO) and
ozone ($O_3$) measurements from the NOAA Nitrogen Oxides and Ozone (NOyO3) instrument
(Bourgeois et al., 2020, 2021) and the merged CO data (from Quantum Cascade Laser System and
NOAA Picarro CO measurements) are used. As we use 2-minute merged ATom measurements,
there are 2796 data points in ATom-2 (January–February 2017) and 3369 data points in ATom-3
(September–October 2017).
**2.4 IAGOS**
The In-service Aircraft for a Global Observing System (IAGOS) is a European research
infrastructure, and was developed for operations on commercial aircraft to monitor atmospheric
composition (Petzold et al., 2015). IAGOS data are available at https://www.iagos.org/iagos-data/
(last access: 3 April 2023). The IAGOS instrument package 1 measures CO, $O_3$, air temperature,
and water vapor (https://www.iagos.org/iagos-core-instruments/package1/). CO is measured by
infrared absorption using the gas filter correlation technique (Precision: ±5%, Accuracy: ±5 ppb)
while $O_3$ is measured by UV absorption at 253.7 nm (Precision: ±2%, Accuracy: ±2 ppb). We use
airborne measurements of CO, $O_3$, air temperature, and water vapor from IAGOS for model
evaluation. The locations of the IAGOS flight tracks over Africa are shown in Figure 1b. The
model results and IAGOS data comparisons are conducted separately for five African sub-regions
(defined in Figure 1b). The IAGOS instruments are onboard commercial airliners and the sampling
may not be representative of the whole sub-regions. For example, IAGOS data over southern
Africa only covers the west part of southern Africa.
**2.5 Ozonesondes**
The ozonesonde is a balloon-borne instrument that measures atmospheric $O_3$ profiles
through the electrochemical concentration cell using iodine/iodide electrode reactions (Thompson
et al., 2017), with records of temperature, pressure, and relative humidity from standard
radiosondes. NASA/GSFC SHADOZ data are available at https://tropo.gsfc.nasa.gov/shadoz/ (last
access: 3 April 2023). We use ozonesonde data from Southern Hemisphere ADditional
OZonesondes (NASA/GSFC SHADOZ; Thompson et al., 2017; Witte et al., 2017, 2018).
Specifically, ozonesonde data from four sites are used (Figure 1b): Ascension (Ascension Island,
U.K.), Nairobi (Kenya), Irene (South Africa), and La Reunion (La Réunion Island, France). The
average $O_3$ measurement uncertainty ranged from 5–9% for the ozonsonde data used in this study.
**2.6 WDCGG**
Monthly surface CO measurements from the World Data Center for Greenhouse Gases
(WDCGG; operated by the Japan Meteorological Agency in collaboration with the World
Meteorological Organization) are used for model evaluation. WDCGG data are available at
https://gaw.kishou.go.jp/ (last access: 3 April 2023). Data from six sites are used (Figure 1b),
namely (Ascension Island, U.K.), Assekrem (Algeria; remote site located in Saharan desert),
Gobabeb (Namibia; located at the base of a linear sand dune, next to an interdune plain), Cape
Point (South Africa; site exposed to the sea on top of a cliff 230 meters above sea level), Izana
(Tenerife, Spain; located on the Island that is ~300 km west of the African coast), and Mare
(Seychelles; near an international airport).
**2.7 Surface PM$_{2.5}$**
At the U.S. embassies, regulatory-grade monitoring data are collected with Beta
Attenuation Monitors (BAMs), using a federal equivalent monitoring method, with an accuracy
within 10% of federal reference methods (Watson et al., 1998; U.S. EPA, 2016). These instruments
are operated by the U.S. State Department and the U.S. EPA, and data are available through
AirNow (https://www.airnow.gov/international/us-embassies-and-consulates/). We use the
measurements at the U.S. embassy locations in Addis Ababa Central (Ethiopia, 9.06° N, 38.76° E)
and Kampala (Uganda, 0.30° N, 32.59° E) for the year 2017 as references (Malings et al., 2020)
to match our simulations. The raw data are made available hourly and for this study we use daily
mean PM$_{2.5}$ for comparison with model simulations. Djossou et al. (2018) presented PM$_{2.5}$
measurements from Feb 2015 to March 2017 at two cities in West Africa – Abidjan and Cotonou
(Figure 1b). In Abidjan, there were three sites that are representative of traffic, waste burning at
landfill, and domestic fires. The site in Cotonou is close to traffic emissions. The concentrations
of PM$_{2.5}$ particles were measured at a weekly time step by the ambient air pumping technique
(Djossou et al., 2018). We compare model results with the weekly PM$_{2.5}$ measurements from the
sites in Abidjan and Cotonou for January–March 2017.

## 2.8 MOPITT


The Measurements of Pollution in the Troposphere (MOPITT) instrument on board the
NASA Terra satellite provides both thermal-infrared (TIR) and near-infrared (NIR) radiance
measurements since March 2000. MOPITT CO data can be accessed through
https://search.earthdata.nasa.gov/search (last access: 3 April 2023). Retrievals of CO column
density and vertical profiles are provided in a multispectral TIR–NIR joint product which has
sensitivity to near-surface as well as free tropospheric CO (Deeter et al., 2011; Worden et al.,
2010). Here we use the MOPITT Version 9 Level 2 CO column product (Deeter et al., 2022) over
Africa to evaluate the MUSICAv0 and WRF-Chem simulations. MOPITT Version 9 has
significant updates to the cloud detection algorithm and NIR calibration scheme. The MOPITT
satellite pixel size is ~22 km × 22 km, and the overpass time is ~10:30 am local time in 2017.
When comparing model outputs to MOPITT the recommended data quality filter is applied and
model outputs are interpolated to the MOPITT retrievals in space and time. To perform
quantitative comparisons, the MOPITT averaging kernel and a priori are used to transform the
model CO profiles to derive model column amounts.

## 2.9 OMI $NO_2$ (QA4ECV)


Tropospheric column $NO_2$ from the Ozone Monitoring Instrument (OMI) on board Aura
is compared to the model in this study. Specifically, the $NO_2$ product from the quality assurance
for the essential climate variables (QA4ECV) project is used (Boersma et al., 2017a; Compernolle
et al., 2020). OMI $NO_2$ data are available at https://www.temis.nl/qa4ecv/no2.html (last access: 3
April 2023). The satellite pixel size is ~13 km × 25 km, and the overpass time is ~1:40 pm local
time in 2017. A data quality filter was applied following the Product Specification Document
(Boersma et al., 2017b; processing_error_flag = 0, solar_zenith_angle < 80, snow_ice_flag < 10
or snow_ice_flag = 255, amf_trop/amf_geo > 0.2, and cloud_radiance_fraction_no20 <= 0.5).
Model profiles were transformed using the provided tropospheric air mass factor (AMF) and
averaging kernels.

## 2.10 OMI HCHO (QA4ECV)


We also use tropospheric column HCHO from OMI in this study. Similar to OMI $NO_2$, we
also use OMI HCHO product from QA4ECV (De Smedt et al., 2017a). OMI HCHO data are
available at https://www.temis.nl/qa4ecv/hcho.html (last access: 3 April 2023). A data quality
filter was applied following the Product User Guide (De Smedt et al., 2017b; processing_error_flag
= 0 and processing_quality_flag = 0). Model profiles were transformed using provided averaging
kernels. We note that HCHO retrievals are subject to relatively large uncertainties compared to
other satellite products used in this study. Therefore, the comparisons between model results and
the OMI HCHO product only indicate the model-satellite discrepancies rather than determining
model deficiencies. In addition, the WRF-Chem simulation from Kumar et al. (2022) does not
include HCHO in the output and hence will not be compared.

## 2.11 MODIS AOD


The aerosol optical depth (AOD) product (550 nm) from the Moderate Resolution Imaging
Spectroradiometer (MODIS) on board Terra NASA Terra satellite is used. MODIS AOD data can
be accessed through https://search.earthdata.nasa.gov/search (last access: 3 April 2023).
Specifically, we used the MODIS Level 2 Collection 6.1 product (MOD04_L2; Levy et al., 2017).
Deep Blue Aerosol retrievals are used (Hsu et al., 2013; Levy et al., 2013) to include retrievals
over the desert. The MODIS satellite pixel size is ~1 km × 1 km, and the overpass time is ~10:30
am local time. East and Southern Africa have complex terrain due to mountains and rift valleys.
This may lead to some uncertainties in MODIS AOD retrievals.
**2.12 AERONET AOD**
We use AOD measurements from the AErosol RObotic NETwork (AERONET; Holben et
al., 1998, 2001). AERONET data can be accessed through https://aeronet.gsfc.nasa.gov/. We use
Level 2 daily data (quality assured), with pre-field and post-field calibration applied and has been
automatically cloud cleared and manually inspected. AOD at 675 nm from AERONET data are
converted to AOD at 550 nm using provided Angstrom exponent to compare with modeled AOD
at 550 nm.
**2.13 SAAQIS**
We also compare model results with $PM_{2.5}$, CO, $NO_2$, and $O_3$ measurements from South
Africa Air Quality Information System (SAAQIS; Gwaze et al., 2018; Tshehla et al., 2019).
SAAQIS is available at http://saaqis.environment.gov.za/. The data are hourly and we calculate
daily average values before compare with model results. Similar to Zhang et al. (2021), we
removed negative values and only calculate daily averages when 75% or more of the hourly data
are available.
**3. Model comparisons with satellite data and evaluation with in situ observations**
Africa includes a wide range of environments and emissions source. Therefore, in this
section we separate the continent in five sub-regions for analysis following Kumar et al. (2022).
CO is a good tracer of anthropogenic and biomass burning emissions and modeled CO tracers are
used in this section to understand sources. CO is a commonly used tracer in models with only one
photochemical sink and an intermediate lifetime (e.g., Tang et al., 2019). CO tracers also allow
clear identification of simulated anthropogenic and biomass burning contributions. Therefore,
tagging CO is computationally efficient and tagged CO is relatively reliable as a tracer in models.
Meteorology has a significant impact on the distributions of pollutants across the regions (e.g.,
Gordon et al., 2023). The CO tracers in the model go through the same model processes (e.g.,
transport) as CO. Therefore, the source contribution shown by the CO tracers is a result of both
emissions and transport. Figure 2 shows the seasonal averages of CO column distributions over
Africa from MOPITT along with the MUSICAv0 and WRF-Chem biases. The highest levels of
CO in these maps are primarily associated with biomass burning, which moves around the
continent with season. Both MUSICAv0 and WRF-Chem simulations underestimate the CO
column compared to MOPITT (Figures 3a and 3b). Overall, MUSICAv0 agrees better with the
OMI tropospheric $NO_2$ column (Figure 3c) and MODIS AOD (Figure 3e) than WRF-Chem
(Figures 3d and 3f). The MUSICAv0 simulation overall has lower tropospheric HCHO column
than OMI in all regions and seasons (Figure 3g). Spatial distributions of model biases against the
OMI tropospheric $NO_2$ column, MODIS AOD, and OMI tropospheric HCHO column are included
in Figures 4 and Figures S1–S2. In this section we compare the model results with satellite data
and in situ observations over sub-regions in Africa and oceans near Africa (Figure 1b). AERONET
data are overlayed with MODIS data in Figure 4. Overall, MODIS and AERONET AOD are
consistent.

## 3.1 North Africa

Over North Africa, both MUSICAv0 and WRF-Chem simulations underestimate the CO column during 2017 (Figures 2 and 3). As shown by the tagged model CO tracers (Figure 5), CO over North Africa is mainly driven by transport of CO from outside the continent and anthropogenic emissions. The model underestimation compared to the MOPITT CO column is consistent with the results of the comparisons with surface CO observations from WDCGG at the two sites located in North Africa (Assekrem and Izana; Figures 6a and 6c). At the two surface sites, the composition of source types and source regions are close to the composition of source types and source regions of the column average over North Africa (Figure 5 and Figures S3 and S4), hence the two sites are representative of the background conditions of North Africa. Compared to MODIS AOD, WRF-Chem has a mean bias of 0.36 whereas MUSICAv0's mean bias is 0.17 for 2017. The model AOD biases over North Africa are likely driven by dust. No comparison is made with IAGOS $O_3$ in North Africa due to data availability.

## 3.2 West Africa

Over West Africa, fire and anthropogenic emissions are both important for CO pollutant and fire impacts peak in DJF (December, January, and February). Compared to the MOPITT CO column, the mean bias of MUSICAv0 and WRF-Chem for West Africa peak around February – the dry season of the Northern Hemisphere (Figure 3). In February, the MUSICAv0 mean bias is $-1.1\times10^{18}$ molecules/cm$^2$ and WRF-Chem mean bias is $-7.5\times10^{17}$ molecules/cm$^2$, which are likely driven by fire emission sources (Figure 5). Model comparisons with IAGOS CO also show a similar bias – both model simulations underestimate CO at all vertical levels. The underestimation peaks during DJF and below 600 hPa (Figure 7). As for MODIS AOD, WRF-Chem has the mean bias 0.69 whereas MUSICAv0's mean bias is 0.15, respectively. Similar to North Africa, the model biases in AOD over West Africa are also likely driven by dust and biomass burning. We also compare modeled $O_3$ with IAGOS $O_3$ observations (Figure 8).

Over West Africa, both models agree well with the IAGOS $O_3$ observations below 800 hPa (mean bias ranges from -1 to -4 ppb). Above 800 hPa over West Africa, WRF-Chem underestimates $O_3$ while MUSICAv0 overestimates $O_3$. Overall, MUSICAv0 consistently overestimates $O_3$ above 800 hPa in all seasons while the direction of WRF-Chem bias changes with seasons (Figure 8). When MUSICAv0 overestimates $O_3$, the bias is in general larger at the higher altitude of the troposphere. The concentration of the model stratospheric ozone tracer, O3S, is also larger at the higher altitude in DJF (Figure 10). The correlation of modeled $O_3$ and O3S is 0.54, and the correlations of $O_3$S and model $O_3$ bias (modeled $O_3$ minus IAGOS $O_3$) is 0.35 over West Africa, implying the overestimation of $O_3$ in the upper troposphere could be partially driven by too strong stratosphere-to-troposphere flux of ozone. Previous studies also found impacts of stratosphere-to-troposphere flux of ozone over West Africa (e.g., Oluleye et al., 2013). Lightning NO emissions can also impact $O_3$ in the upper troposphere. The MUSICAv0 simulation has somewhat (~3 times) higher lightning NO emissions (Figure S5) compared to a standard CAM-chem simulation (not shown), therefore the high ozone in the upper troposphere may be due to an over-estimate of lightning NO. We also compared our modeled lightning NO emissions with a multi-year average climatology (2008-2015) from Maseko et al. (2021) over South Africa, and found that the seasonal cycle from MUSICAv0 and standard CAM-chem are consistent with the climatology. The magnitude of MUSICAv0 lightning NO emissions overall agree better with the climatology compared to that from standard CAM-chem simulation. Impacts of lightning NO emissions on upper troposphere $O_3$ in MUSICAv0 will be investigated and evaluated further in the

future. A brief comparison with IAGOS measurements of air temperature and water vapor profiles
over West Africa as well as other sub-regions shows that MUSICAv0 overall agrees well with
these meteorological variables (Figure S6).
We compare the models with weekly $PM_{2.5}$ measurements at 3 sites in Abidjan
(representing domestic fires emissions, waste burning at landfill, and traffic) and 1 site in Cotonou
representing traffic emissions (Figure S7). Overall, both models underestimate $PM_{2.5}$ at the three
Abidjan sites, especially near the domestic fire emissions where measured $PM_{2.5}$ exceeded 400
$\mu g/m^3$. We include open burning emissions in the MUSICAv0 simulation however the significant
underestimation point to the possibility of missing emissions. Moreover, these three sites in
Abidjan are within the same city and near strong emission sources and hence are challenging for
both models to resolve. In fact, they fall into the same model grids and therefore model values at
the three sites are the same for both models. This demonstrates the need of higher model resolution
to resolve variabilities of air quality in a city.
**3.3 Central Africa**
Compared to MOPITT CO column, the mean bias of MUSICAv0 and WRF-Chem for
Central Africa varies with seasons (Figure 3) but peaks during the dry season in September
(MUSICAv0 mean bias of $-1.0 \times 10^{18}$ molecules/cm$^2$; WRF-Chem mean bias of $-1.2 \times 10^{18}$
molecules/cm$^2$). The tagged model CO tracers show that in September, local fire emissions are the
dominant driver of CO in Central Africa (Figure 5). Compared to the IAGOS CO profiles (Figure
7), both models have the largest bias over Central Africa among the sub-regions in Africa – mean
bias of MUSICAv0 and WRF-Chem are -46 ppb and -36 ppb, respectively. The high bias over
Central Africa mainly occurs during the fire season. In central Africa, both models also
underestimate $NO_2$ (mean biases of MUSICAv0 and WRF-Chem are $-1.5 \times 10^{14}$ and $-5.5 \times 10^{14}$
molecules/cm$^2$, respectively). The underestimations in both CO and $NO_2$ by the two model
simulations are likely driven by the underestimation in fire emissions. Indeed, the emission
estimates from the newest version of FINN (FINNv2.5; Wiedinmyer et al., 2023) are higher
compared to both QFED (used in the MUSICAv0 simulation) and FINNv1.5 (used in the WRF-
Chem simulation) in this region.
Model mean bias of HCHO ($-1.3 \times 10^{16}$ molecules/cm$^2$ for the whole 2017) over Central
Africa is the largest among the five regions (Figure 3). The spatial distribution of HCHO bias
(Figure S2) largely co-locates with the vegetation (Figure 9). Over the barren or sparsely vegetated
area in North Africa, HCHO biases are relatively small while over the vegetated area HCHO bias
are relatively large. Over North Africa, the mean bias is $-0.66 \times 10^{16}$ molecules/cm$^2$ for the whole
2017 whereas over the other four regions, the mean bias ranges from $-0.93 \times 10^{16}$ molecules/cm$^2$ to
$-1.31 \times 10^{16}$ molecules/cm$^2$ for the whole 2017. This indicates that the negative bias in MUSICAv0
HCHO could be due to underestimated biogenic emissions in the model. In addition, the
underestimation of HCHO in Central Africa (Figure S2) co-locates with the underestimation of
CO in time and space (Figure S1), implying that fire emissions that contributed to model CO biases
may also contribute to the HCHO underestimation in MUSICAv0 during fire season. It is
important to note that the uncertainty of OMI tropospheric HCHO column is relatively large
compared to other satellite products. Here the averaged retrieval uncertainty (random and
systematic) is ~120%.
When compared to the IAGOS $O_3$ profiles over Central Africa (Figure 8), both models
agree well with the IAGOS $O_3$ observations below 800 hPa (mean bias ranges from -1 to -4 ppb).
Above 800 hPa, WRF-Chem underestimates $O_3$ while MUSICAv0 overestimates $O_3$. The
correlation of modeled $O_3$ and O3S is 0.67, and the correlations of O3S and model $O_3$ bias is 0.50
over Central Africa, indicating $O_3$ overestimation in Central Africa are more likely to be impacted
by stratosphere-to-troposphere flux of ozone than that in West Africa.
**3.4 East Africa**
CO over East Africa is dominated by local emissions and inflow from outside the continent.
Fire and anthropogenic emissions contribute approximately the same to CO over East Africa
(Figure 5). Both MUSICAv0 and WRF-Chem simulations underestimate the CO column
compared to MOPITT (Figure 3), and the WRF-Chem simulation also underestimate the
tropospheric $NO_2$ column compared to OMI. The biases in CO column and tropospheric $NO_2$
column peak in September. One possible driver could be fire emissions from other regions (Figure
5), however, further studies will be needed to address this.
Compared to IAGOS $O_3$ profiles over East Africa, biases of MUSICAv0 below 600 hPa
has a seasonal variation while over 600 hPa are consistently positive (Figure 8). The correlations
of O3S and model $O_3$ bias against IAGOS data is 0.50 in the region. The correlations between O3S
and model $O_3$ bias are highest over Central and East Africa compared to other regions, indicating
stratosphere influence are strongest in these two regions among the sub-regions. Central and East
Africa are relatively more mountainous therefore topography driven stratospheric intrusions might
be expected. The Nairobi ozonesonde site is located in East Africa (Figure 1b). When comparing
to the $O_3$ profiles from ozonesondes (Figure 10), MUSICAv0 overall overestimates $O_3$ in the
troposphere at the four sites while WRF-Chem tends to underestimate $O_3$ in the free troposphere
(below 200 hPa). The Nairobi site is an exception where both MUSICAv0 and WRF-Chem
simulations significantly overestimate $O_3$ in all seasons (mean bias of MUSICAv0 and WRF-
Chem below 200 hPa are 27 ppb and 20 ppb, respectively). Among the four ozonesonde sites,
correlations of model bias of $O_3$ and O3S are highest at the Nairobi site (0.74) where the model
significantly overestimates $O_3$. The results of model-ozonesonde comparisons are consistent with
the results of model-IAGOS comparisons and indicate a potential issue in modeled stratosphere-
to-troposphere flux of ozone.
We compare the model results with $PM_{2.5}$ measurements from two surface sites in East
Africa (Addis Ababa and Kampala; Figure 1b). Despite using different aerosol methods and
emission inventories, both MUSICAv0 and WRF-Chem underestimate surface $PM_{2.5}$ when
compared to observations at the two sites (Figure 11). The errors in $PM_{2.5}$ concentrations at the
U.S. Embassy in Kampala are especially prominent. However, both models approximate the
variation of the $PM_{2.5}$ in both locations. Many factors contribute to the inconsistency in the
magnitude of modeled $PM_{2.5}$ concentrations. For instance, emission inventories in this region
require additional improvement. In Uganda, increasing motor vehicle ownership and burning
biomass for domestic energy use contribute to ambient $PM_{2.5}$ levels (Clarke et al., 2022; Petkova
et al., 2013; Kinney et al., 2011). Detailed $PM_{2.5}$ composition measurements would also help to
pinpoint the cause of inaccuracies (Kalisa et al., 2018). Model resolutions could also be a potential
reason for the underestimation. Over Kampala, high spatial variability of $PM_{2.5}$ over the urban
environment can contribute to model bias (Atuhaire et al., 2022), as also shown by the AirQo low-
cost air quality monitors (Sserunjogi et al., 2022; Okure et al., 2022).
**3.5 Southern Africa**
Among the five regions, MUSICAv0 has the lowest mean bias in CO ($-3.2 \times 10^{17}$
molecules/cm$^2$ annually) over Southern Africa (Figure 3). WRF-Chem also has low mean bias and
RMSE in CO over Southern Africa except for the months of September, October, and November
(SON) period where WRF-Chem has larger CO mean bias ($-6.2 \times 10^{17}$ molecules/cm$^2$) than
MUSICAv0. Tagged model CO tracers indicate that CO over Southern Africa is significantly
impacted by CO emissions from Central Africa, East Africa, Southern Africa, and inflow from
outside the continent. As for the source types, anthropogenic and fire emissions are both important
and fire impacts peak in September (e.g., Archibald et al., 2009, 2010; Archibald 2016). There are
two WDCGG sites located in Southern Africa (Figure 1b; Gobabeb and Cape Point). When
compared to surface CO observations from WDCGG, both models consistently underestimate CO
by up to 40% at most sites. The Cape Point site in Southern Africa is an exception (Figure 6) where
MUSICAv0 overestimates CO by 40 ppb (annual mean; and up to 78 ppb in May 2017). CO tracers
in the model (Figures S3 and S4) show that the simulated CO at Cape Point is mainly driven by
anthropogenic CO emissions from Southern Africa. Therefore, the overestimation of CO at Cape
Point by MUSICAv0 may be due to an overestimation of emissions in South Africa. Note that the
Cape Point measurement site is located on the tip of southern Africa and has a strong impact from
clean marine air (Labuschagne et al., 2018), which the model likely cannot represent accurately.
As for NO$_2$, WRF-Chem underestimates tropospheric NO$_2$ column in most regions except
for Southern Africa (Figure 3). Over Southern Africa, WRF-Chem overestimates NO$_2$ especially
during June, July, and August (JJA). MUSICAv0 also tends to overestimates NO$_2$ at the same
location in JJA however the bias is not as large as for WRF-Chem.
MUSICAv0 simulation overall has a lower mean bias (0.14 annually) than the WRF-Chem
simulation (mean bias of 0.31 annually) compared to MODIS AOD with Southern Africa being
the only exception (Figure 3). Over Southern Africa, MUSICAv0 overestimates AOD by ~0.21
annually (Figure 3) and the bias peaks in January (mean bias=0.45). This overestimation in AOD
over Southern Africa is not seen in WRF-Chem. It is likely that the MUSICAv0 overestimation in
AOD over Southern Africa is also due to biases in modeled dust as the AOD bias is co-located
with the only barren or sparsely vegetated area in Southern Africa (Figure 9 and Figure S2).
Over Southern Africa, MUSICAv0 tends to overestimate O$_3$ compared to IAGOS at all
levels at all seasons in 2017 (Figure 8). The MUSICAv0 O$_3$ bias is 5-10 ppb below 800 hPa for
the four seasons and 23-39 ppb at 225 hPa. The concentration of O3S over Southern Africa is
higher than those over other regions. However, the correlation of O$_3$S and model O$_3$ bias is lower
than other regions (0.13) indicating stratosphere-to-troposphere flux of ozone may not be the main
driver of O$_3$ bias over Southern Africa even though stratosphere-to-troposphere flux of ozone are
relatively strong in the region (e.g., Leclair De Bellevue et al., 2006; Clain et al., 2009; Mkololo
et al., 2020). The Irene ozonesonde site is located in Southern Africa (Figure 1b). Compared to the
ozonesonde O$_3$ profiles at the Irene site, however, the sign of MUSICAv0 has a seasonal variation
(Figure 10e-10h). For example, at 675–725 hPa, MUSICAv0 O$_3$ bias in MAM and JJA is 3-9 ppb
whereas in SON and DJF it is -2 to -6 ppb. The IAGOS measurements and the Irene ozonesonde
site are not co-located, so the difference is expected due to the different sampling locations and
environment. Compared to other ozonesonde sites, the correlation of O$_3$S and model O$_3$ bias over
Southern Africa is lower (0.14) and MUSICAv0 agrees relatively well with observations, which
is consistent with the comparison results with IAGOS data (Figure 8).
We further compare MUSICAv0 and WRF-Chem results with surface PM$_{2.5}$, CO, NO$_2$,
and O$_3$ measurements from SAAQIS in South Africa (Figures S8-S11). Overall, the performance
of MUSICAv0 and WRF-Chem compared to SAAQIS data are similar. Both models underestimate
surface CO in most sites (consistent with the comparisons with satellites) with exceptions near
Gauteng (industrialized and urbanized region). Compared to SAAQIS sites near Cape Point,
MUSICAv0 does not show overestimation which is opposite to the overestimation compared to
WDCGG Cape Point site. The maximum value of monthly CO observations from WDCGG Cape
Point site in 2017 is ~150 ppb whereas the seasonal mean values of SAAQIS CO measurements
near Cape Point site can be up to 600 ppb. SAAQIS CO measurements near Cape Point shows
relatively large spatial variability, indicating (1) that there may be a wide range of emission sources
that are poorly captured by the model and (2) a large role of local sources and potentially complex
meteorology. In addition, uncertainties in observations could also contribute to the difference. Both
models tend to overestimate $NO_2$ near Gauteng, which may be related to local emissions. Both
models can either overestimate or underestimate $PM_{2.5}$ and/or $O_3$ at different SAAQIS sites. The
model bias in $PM_{2.5}$ and $O_3$ shows large spatial variability especially near Gauteng. Higher model
resolution is needed to address the highly complex and diverse environment in the region. Lastly,
it is worth pointing out that in South Africa, both models have evident bias in $PM_{2.5}$ near Gauteng
(Figure S11) however modeled AOD from both models agree relatively well with MODIS and
AERONET (Figure 4). More studies are needed to understand this feature.
**3.6 Oceans near Africa**
We compare the CO, NO, and $O_3$ from the MUSICAv0 simulation with measurements
from ATom-2 and ATom-3 in 2017 (Figure 1a) to provide a global benchmark. Measurements
made over the Atlantic Ocean and Pacific Ocean, and in January-February (Jan-Feb) and
September-October (Sep-Oct) are compared separately (Figures 11 and 12). The comparison was
made with data averaged into 10° latitude and 200 hPa bins. Overall, the model consistently
underestimates CO globally in both seasons. The underestimation of CO is a common issue in
atmospheric chemistry models and could be due to various reasons, including emissions,
deposition, and chemistry (e.g., Fisher et al., 2017; Shindell et al., 2006; Stein et al., 2014; Tilmes
et al., 2015; Tang et al., 2018; Gaubert et al., 2020). Specifically for our MUSICAv0 simulation
in this study, the model bias in CO is relatively large (up to 52 ppb) over the Northern Hemisphere
(especially at high latitude and near the surface) and small over the Southern Hemisphere (Figures
11 and 12). Over the Atlantic Ocean, the bias in CO is larger in September-October than Jan-Feb
in both the Northern Hemisphere (-30 ppb in Jan-Feb versus -34 ppb in Sep-Oct) and Southern
Hemisphere (-11 ppb in Jan-Feb versus -14 ppb in Sep-Oct). Over the Pacific Ocean, however, the
CO bias is similar for both time periods in the Northern Hemisphere (-30 ppb) while in the
Southern Hemisphere, the CO bias changes significantly from -8 ppb in Jan-Feb to -16 ppb in Sep-
Oct. The changes in CO bias over the Southern Hemisphere are likely due to seasonal change in
fire emissions. Overall, the mean biases (Figures 11 and 12) suggest that the simulation agrees
better with ATom observations in the Southern Hemisphere than in the Northern Hemisphere, and
in Jan-Feb than in Sep-Oct (Figures 11 and 12), consistent with Gaubert et al. (2016).
In both seasons and both hemispheres, the model in general overestimates $O_3$ in the
stratosphere/UTLS (upper troposphere and lower stratosphere) by up to 38 ppb (above 200 hPa).
In the troposphere (below 200 hPa), the model overall agrees well with the ATom data over the
Pacific Ocean in the Southern Hemisphere (in most cases the bias is less than ±5 ppb). However,
over the Atlantic Ocean in the Southern Hemisphere, MUSICAv0 tends to overestimate $O_3$,
especially in Jan-Feb. In the troposphere of the Northern Hemisphere, MUSICAv0 consistently
overestimates $O_3$ over both oceans and both seasons. The positive bias in $O_3$ decreases from the
upper troposphere towards the surface, indicating that the overestimation of $O_3$ in the troposphere
may be due to stratosphere-to-troposphere flux of ozone. This was also noted for other global
models (Bourgeois et al. 2021). Thompson et al. (2014) found $O_3$ at the Irene site is also influenced
by long-range transport of growing pollution in the Southern Hemisphere, which could also
contribute to the model bias. As for NO, the model tends to overestimate NO above 200 hPa
(approximately the stratosphere and Upper Troposphere-Lower Stratosphere; UTLS) by up to 50
ppt. Overall, the NO biases can be either positive or negative depending on location and season.
The distributions of NO bias (Figures 11 and 12) do not show an overall spatial pattern, unlike
those for CO (which changes monotonically with latitude) or $O_3$ (which changes monotonically
with altitude).

**4. Model application: identifying key regions in Africa for future in situ observations and**
**field campaign(s)**
As a demonstration of the application of MUSICAv0, here we use the results of model-
satellite comparisons to identify potential regions where the atmospheric chemistry models need
to be improved substantially. More field campaigns and more in situ observations would not only
provide observational benchmark dataset to understand and improve the modeling capability in
the region, but would be also useful for the validation and calibration of satellite products. Here
we use Taylor score to quantify model-satellite discrepancies. Taylor score (Taylor, 2001) is
defined by
$$S = \frac{4(1+R)}{(\hat{\sigma}_f + 1/\hat{\sigma}_f)^2 (1+R_0)}$$

where $\hat{\sigma}_f$ is the ratio of $\sigma_f$ (standard deviation of the model) and $\sigma_r$ (standard deviation of
observations), R is correlation between model and observations, and $R_0$ is the maximum
potentially realizable correlation (=1 in this study). Taylor score ranges from 0 to 1 and a higher
Taylor score indicates better satellite-model agreement. To identify potential locations, we
separate the Africa continent into $5° \times 5°$ (latitude × longitude) pixels as shown in Figure 14. And
for each pixel, we calculate Taylor scores of MUSICAv0 compared to the three satellite Level 2
products (e.g., MOPITT CO column retrievals, OMI tropospheric $NO_2$ column retrievals, and
MODIS AOD) separately. Then three Taylor scores are summed up to obtain the total Taylor score
for MUSICAv0 (ranges from 0 to 3) as shown in Figures 13a-13e. A similar calculation is
conducted for WRF-Chem (Figures 13f-13j). Note that we did not include Taylor scores for HCHO
in the total Taylor score due to that (1) WRF-Chem simulations did not save HCHO output, and
(2) the HCHO retrievals have relatively high uncertainties (Taylor scores of MUSICAv0 compared
to OMI tropospheric HCHO column retrievals are provided separately in Figure S12).
Overall, both MUSICAv0 and WRF-Chem have low total Taylor scores in the 30°E – 45°E,
5°S – 5°N region in East Africa (a region of 15° longitude × 10° latitude) during MAM (March,
April, and May), JJA (June, July, and August), and SON (September, October, and November), as
highlighted in Figure 14, indicating relatively large model-satellite discrepancies in the region.
Besides the 30°E – 45°E, 5°S – 5°N region highlighted in Figure 14, there are a few other regions
with low Taylor scores for both MUSICAv0 and WRF-Chem such as 10°E – 20°E, -30°S – -20°N
region and the east of Madagascar.
The 30°E – 45°E, 5°S – 5°N region (a sub-region in East Africa) is also the region where
the Nairobi ozonesonde site and the Kampala surface $PM_{2.5}$ site are located (Figure 1b). As
discussed above, both MUSICAv0 and WRF-Chem significantly overestimate $O_3$ (Figure 10) and
largely underestimate $PM_{2.5}$ (Figure 11) in the region. More in situ observations or future field
campaigns in the region can substantially help in the understanding model-satellite and model-in
situ observation discrepancies and improving model performance.
The 30°E – 45°E, 5°S – 5°N region (a sub-region in East Africa) is potentially a favorable
location for future field campaign(s) not only because of the large model-satellite and model-in
situ observation discrepancies, but also due to that the population density is high and landcover
are diverse in the region (Figure 9). The relatively high population density in the region indicates
that improved air quality modeling in the region can benefit a large population. A diverse landcover
indicates more processes/environments can be sampled. CO tracers in the model (Figure 15) show
that CO over the region is mainly driven by both anthropogenic and fire emissions. Anthropogenic
emissions play a more important role in the 30°E – 45°E, 5°S – 5°N region compared to East
Africa in general (Figures 4 and 14). In terms of source regions, emissions from East Africa and
inflow from outside the continent are the dominant source, with some contributions from Central
Africa. Note that the source analyses using model tracers may be subject to uncertainties in the
emission inventories, in this case CAMSv5.1, QFED, and the waste burning inventory used here.
As discussed above (e.g., Section 3.4), there might be missing sources in the region. In addition,
emission factors used in many emission inventories are based on measurements outside the
continent of Africa (e.g., Lamarque et al. 2010; Klimont et al., 2013; Pokhrel et al. 2021). It is not
clear so far if these emission factors are applicable to emissions in Africa (e.g., Keita et al., 2018).
Therefore, a field campaign in the region can help address these issues.
We would like to point out that in this analysis, the key area is selected using 3 satellite
products/chemical species and two models. The Taylor score is a comprehensive measure of model
performance that accounts for variance and correlation, however, other models and types of
comparisons may provide different answers.
**5. Conclusions**
Africa is one of the most rapidly changing regions in the world and air pollution is a
growing issue at multiple scales over the continent. MUSICAv0 is a new community modeling
infrastructure that enables the study of atmospheric composition and chemistry across all relevant
scales. We developed a MUSICAv0 grid with Africa refinement (~28 km × 28 km over Africa and
~110 km × 110 km for the rest of the world) and conducted the simulation for the year 2017. We
evaluated the model with in situ observations including ATom-2 and ATom-3 airborne
measurements of CO, NO, and $O_3$, IAGOS airborne measurements of CO and $O_3$, $O_3$ profiles from
ozonesondes, surface CO observations from WDGCC, and surface $PM_{2.5}$ observations from two
U.S. Embassy locations. We then compare MUSICAv0 with satellite products over Africa, namely
MOPITT CO column, MODIS AOD, OMI tropospheric $NO_2$ column, and OMI tropospheric
HCHO column. Results from a WRF-Chem simulation were also included in the evaluations and
comparisons as a reference. Lastly, as an application of the model, we identified potential African
regions for in situ observations and field campaign(s) based on model-satellite discrepancies
(quantified by Taylor score), with regard to model-in situ observation discrepancies, source
analyses, population, and land cover. The main conclusions are as follows.
(1) When comparing to ATom-2 and ATom-3, MUSICAv0 consistently underestimates
CO globally. Overall, the negative model bias increases with latitude from the Southern
Hemisphere to the Northern Hemisphere. MUSICAv0 also tends to overestimate $O_3$ in the
stratosphere/UTLS, and the positive model bias overall decreases with altitude.

(2) The MUSICAv0 biases in $O_3$ when compared to ATom, IAGOS, and ozonesondes are
likely driven by stratosphere-to-troposphere fluxes of $O_3$ and lightning NO emissions.

683   (3) Overall, the performance of MUSICAv0 and WRF-Chem are similar when compared
684    to the surface CO observations from six WDCGG sites in Africa.

685   (4) Both models have negative bias compared to the MOPITT CO column, especially over
686    Central Africa in September, which is likely driven by fires.

687   (5) Overall, MUSICAv0 agrees better with OMI tropospheric $NO_2$ column than WRF-
688    Chem.

689   (6) MUSICAv0 overall has a lower tropospheric HCHO column than OMI retrievals in all
690    regions and seasons. Biogenic and fire emissions are likely to be the main driver of this
691    disagreement.

692   (7) Over Africa, the MUSICAv0 simulation has smaller mean bias and RMSE compared
693    to MODIS AOD than the WRF-Chem simulation.

694   (8) The 30°E – 45°E, 5°S – 5°N region in East Africa is potentially a favorable location for
695    future field campaign(s) not only because of the large model-satellite and model-in situ
696    observation discrepancies, but also due to the population density, landcover, and pollution
697    source in this region.

698   Overall, the performance of MUSICAv0 is comparable to WRF-Chem. The
699 underestimation of CO is a common issue in atmospheric chemistry models such as MUSICAv0
700 and WRF-Chem. The overestimation of $O_3$ in MUSICAv0 is likely driven by too strong of
701 stratosphere-to-troposphere fluxes of $O_3$ and perhaps an over-estimate of lightning NO emissions,
702 however, future studies are needed to confirm and solve this issue. The significant underestimation
703 in surface $PM_{2.5}$ at two sites in East Africa and the overall overestimation in AOD in Africa
704 compared to MODIS imply missing local sources and an overestimation of dust emissions, and
705 require further study. In addition, lack of data could also contribute to disagreement in model and
706 in situ observations as one site in a city is not representative of the full city. Field campaigns and
707 more in situ observations in 30°E–45°E, 5°S–5°N region in East Africa (as well as other regions
708 in Africa) are necessary for the improvement of atmospheric chemistry model(s) as shown by the
709 MUSICAv0 and WRF-Chem simulations.

710   Fire and dust are important sources of air pollution in Africa. The performance of
711 MUSICAv0 is degraded during fire season and over dust regions. Uncertainties in emission
712 estimates of fire and dust and in the model representation of atmospheric processes could
713 potentially contribute to the model biases. Future studies on fire and dust in Africa are needed to
714 address these uncertainties and air quality modeling over Africa.

715   Here we divided the continent into five sub-regions to show the overall performance of
716 MUSICAv0 over sub-regions of Africa. This accounted for the diversity in atmospheric chemistry
717 environment to some degree. However, each sub-region is not homogeneous. In fact, different
718 cities in the same sub-region may have different emission characteristics. In the future when
719 specific scientific questions are studied with MUSICAv0, we will use higher resolution to address
720 the highly complex and diverse environment. We plan to conduct a model simulation for multiple
721 years and develop additional model grids with potentially higher resolution in Africa sub-regions
722 based on the current MUSICAv0 Africa grid. Higher resolution will benefit the comparisons of
723 model and in situ observations. The future simulation will be conducted for years after 2017 as
724 there are more in situ observations available in recent years.

725

**Code and data availability**

The model code used here can be accessed through https://doi.org/10.5281/zenodo.8051435. The data produced by this study can be accessed through https://doi.org/10.5281/zenodo.8051443.

**Acknowledgement**

This material is based upon work partially supported by the National Aeronautics and Space Administration under Grant No. 80NSSC23K0181 issued through the NASA Applied Sciences SERVIR program. We thank ATom, WDCGG, IAGOS, NASA/GSFC SHADOZ teams, and the U.S. State Department and the U.S. EPA for in situ observations. We thank Anne Thompson and Gonzague Romanens for detailed explanation of SHADOZ Ozonesonde data format. We thank MOPITT, MODIS AOD, OMI $NO_2$ and OMI HCHO teams for the satellite products. The NCAR MOPITT project is supported by the National Aeronautics and Space Administration (NASA) Earth Observing System (EOS) program. We thank the QA4ECV project. We thank Sabine Darras for CAMSv5.1 emissions. We would like to acknowledge high-performance computing support from Cheyenne (doi:10.5065/D6RX99HX) provided by NCAR's Computational and Information Systems Laboratory, sponsored by the National Science Foundation. This material is based upon work supported by the National Center for Atmospheric Research, which is a major facility sponsored by the National Science Foundation under Cooperative Agreement No. 1852977. We thank James Hannigan, Ivan Ortega, Siyuan Wang, and all the attendees of ACOM CAM-chem/MUSICA weekly meeting for helpful discussions.

**Competing interests**

The contact author has declared that neither they nor their co-authors have any competing interests.

**Author contributions**

WT, LKE, HMW, and PL were involved in the initial design of this study. WT led the analysis. RK and CH conducted the WRF-Chem simulation. ZZ interpretated $PM_{2.5}$ results. BG, ST, SM and other coauthors provide discussions. RRB helped with QFED emissions. CG and AS produced CAMSv5.1 emissions. KM, BCD, JP, and CT conducted measurements during ATom. WT prepared the paper with improvements from all coauthors.

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

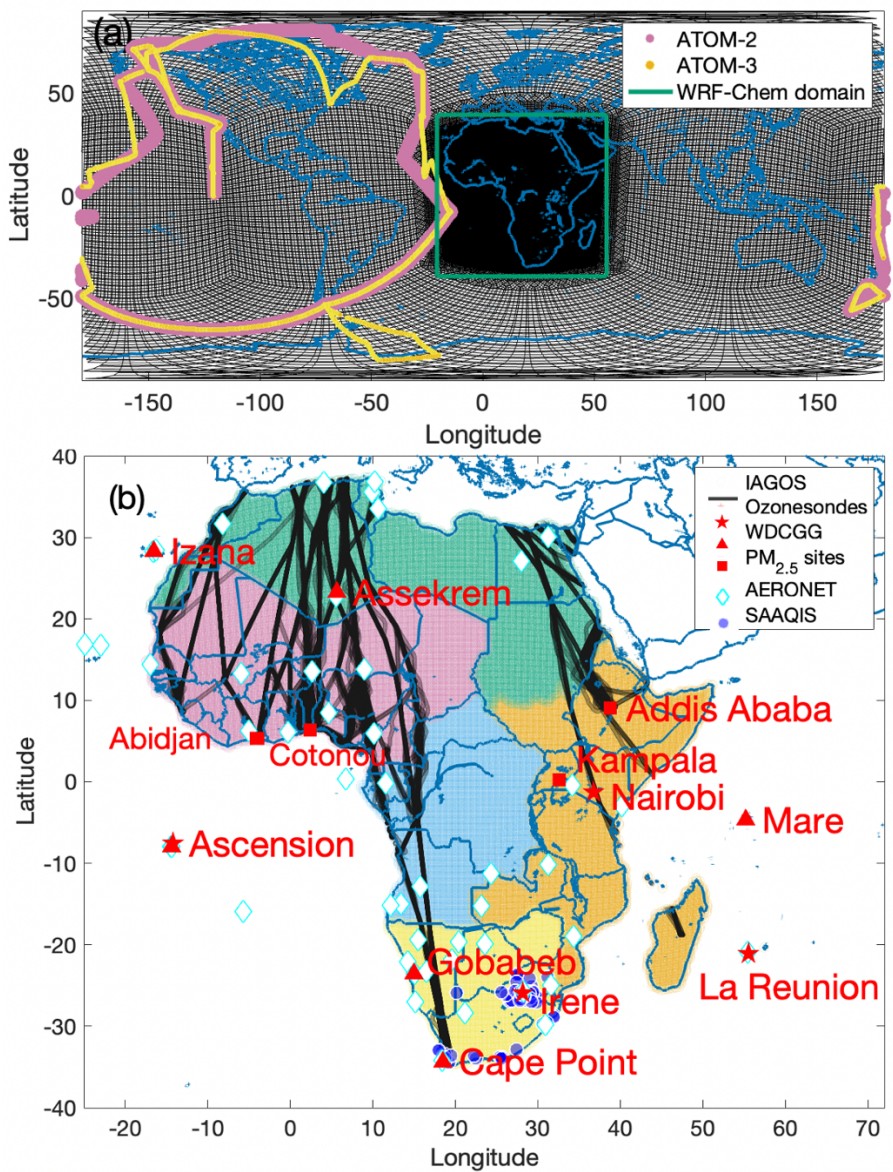

**Figure 1.** Model grid, in situ observations used in this study, and sub-regions in Africa. (a)
MUSICAv0 model grid developed for Africa in this study (black), domain boundary of the WRF-
Chem simulation compared in this study (shown by green box), observations from the
Atmospheric Tomography Mission (ATom) field campaign 2 (ATom-2; 2017 Jan to 2017 Feb;
pink) and ATom-3 (2017 Sep to 2017 Oct; yellow). (b) Sub-regions in Africa are shown, namely
North Africa (green), West Africa (pink), East Africa (orange), Central Africa (blue), and Southern
Africa (yellow). Location of in situ observations are labeled on the map. Flight tracks of the In-
service Aircraft for a Global Observing System (IAGOS) are shown with black lines. Four
ozonesonde sites are shown by pentagrams (Ascension, Irene, Nairobi, and La Reunion); six sites
from the World Data Centre for Greenhouse Gases are shown by triangles (Assekrem, Cape Point,
Izana, Gobabeb, Mare, and Ascension); surface sites for $PM_{2.5}$ are shown by squares (Addis Ababa
and Kampala in East Africa; Abidjan and Cotonou in West Africa); AErosol RObotic NETwork
(AERONET) sites are shown with diamond; South Africa Air Quality Information System
(SAAQIS) sites are shown with blue circles.

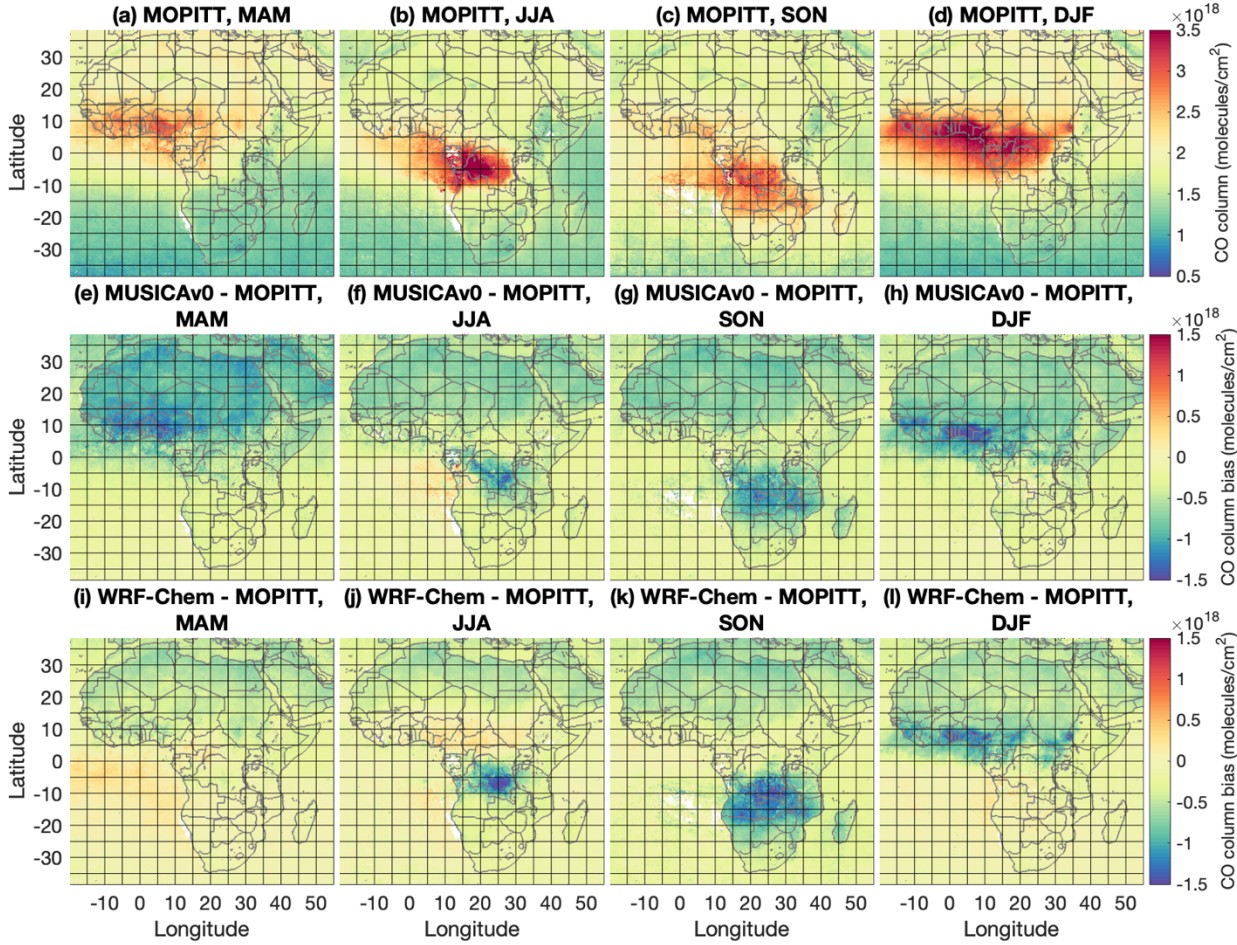

**Figure 2**. Comparisons of MUSICAv0 and WRF-Chem simulations to MOPITT CO column
(molecules/cm²) for each season of 2017. (a-d) Averaged MOPITT CO column: MAM (March,
April, and May), JJA (June, July, and August), SON (September, October, and November), and
DJF (December, January, and February). (e-h) MUSICAv0 model biases against MOPITT CO
column for MAM, JJA, SON, and DJF. (i-l) is the same as (e-h) but for WRF-Chem. All data are
gridded to 0.25 degree × 0.25 degree for plotting.

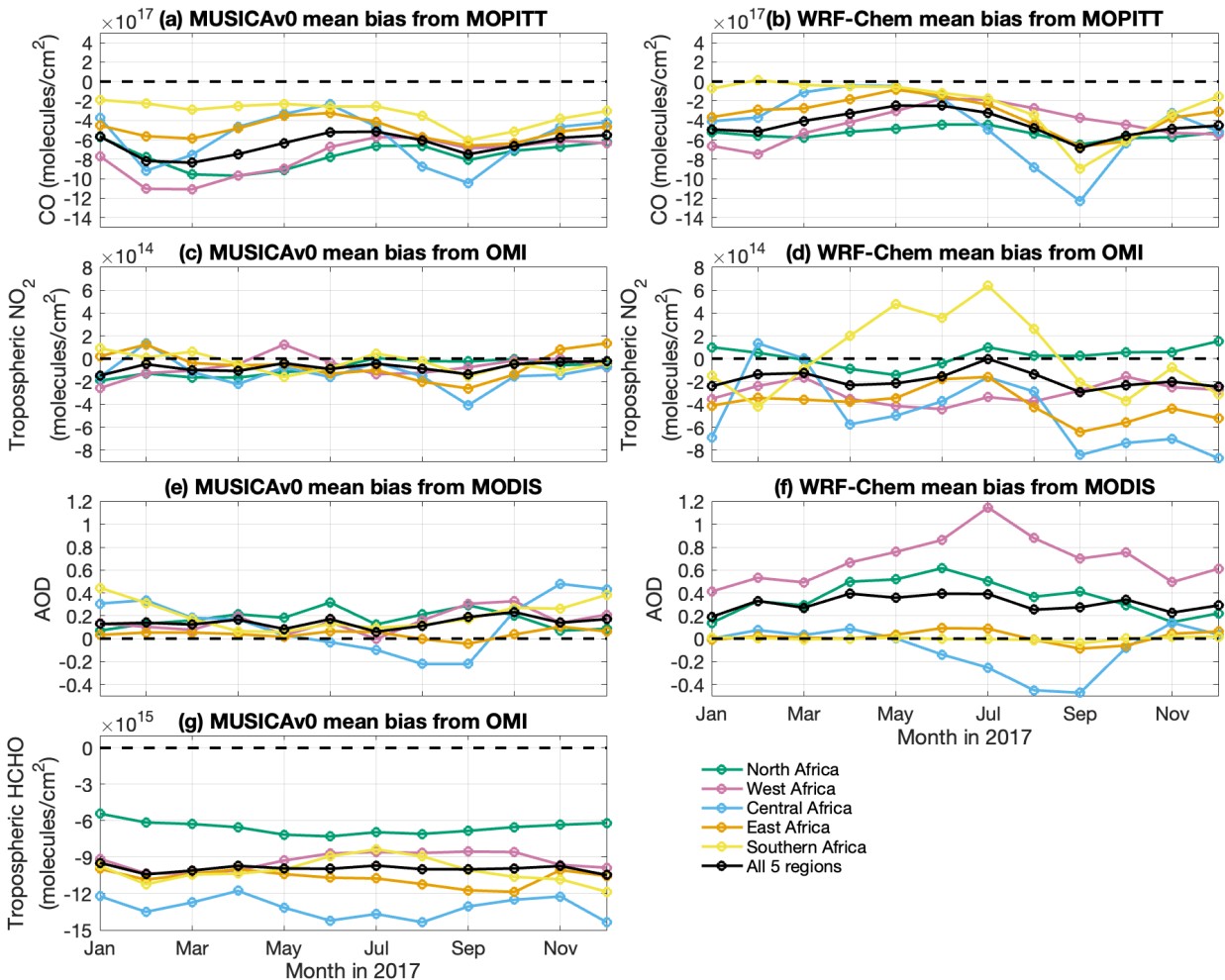

**Figure 3**. Mean bias of MUSICAv0 and WRF-Chem simulations from satellite data. Monthly timeseries of mean bias of (a) MUSICAv0 and (b) WRF-Chem against MOPITT CO column (molecules/cm²) in 2017 over Africa (black), North Africa (green), West Africa (pink), East Africa (orange), Central Africa (blue), and Southern Africa (yellow). (c-d) are same as (a-b) but for mean bias against OMI tropospheric $NO_2$ column (molecules/cm²). (e-f) are same as (a-b) but for mean bias against with MODIS (Terra) Aerosol Optical Depth (AOD). (g) is the same as (a) but for mean bias against OMI tropospheric HCHO column (molecules/cm²).

1225

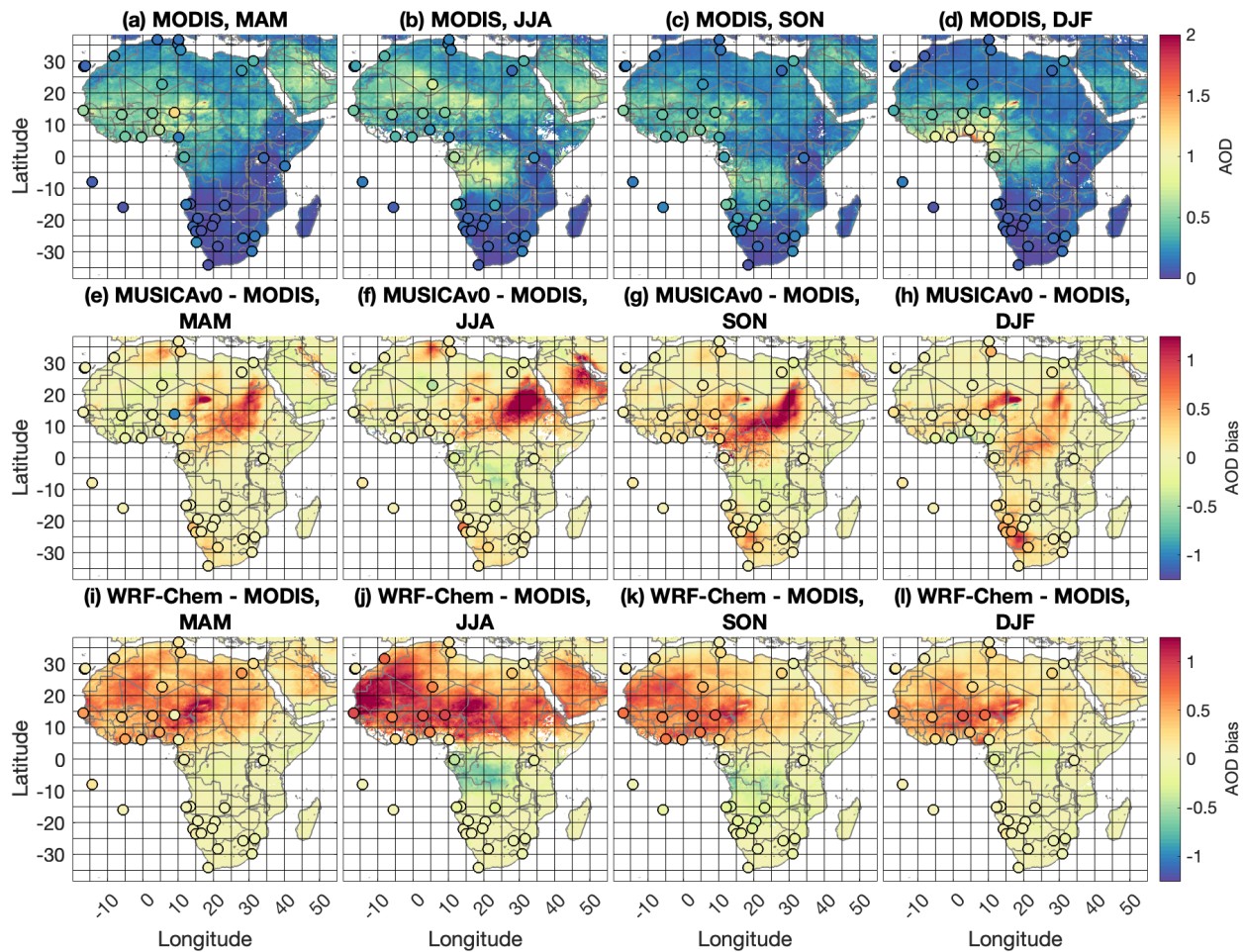

**Figure 4**. Comparisons of MUSICAv0 and WRF-Chem simulations and MODIS and AERONET AOD at 550 nm in 2017. (a-d) Averaged MODIS and AERONET AOD in MAM (March, April, and May), JJA (June, July, and August), SON (September, October, and November), and DJF (December, January, and February). (e-h) MUSICAv0 model biases against MODIS and AERONET AOD in MAM, JJA, SON, and DJF. (i-l) is the same as (e-h) but for WRF-Chem. All data are gridded to 0.25 degree × 0.25 degree for plotting. AERONET AOD in (a-d) and model bias against AERONET AOD in (e-l) are shown by the circles overlayed on the map.

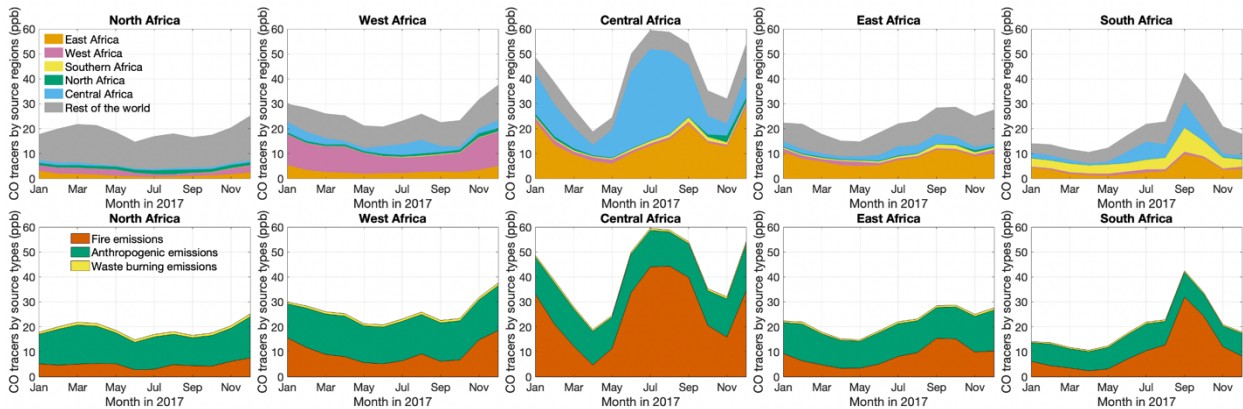

**Figure 5.** Monthly time series of column-averaged CO tracers in North Africa, West Africa, East
Africa, Central Africa, and Southern Africa. Top panels show CO tracers of emissions from North
Africa (green), West Africa (pink), East Africa (orange), Central Africa (blue), Southern Africa
(yellow), and the rest of the world (grey). Bottom panels show CO tracers of fire emissions (red),
anthropogenic emissions (green), and waste burning emissions (yellow).

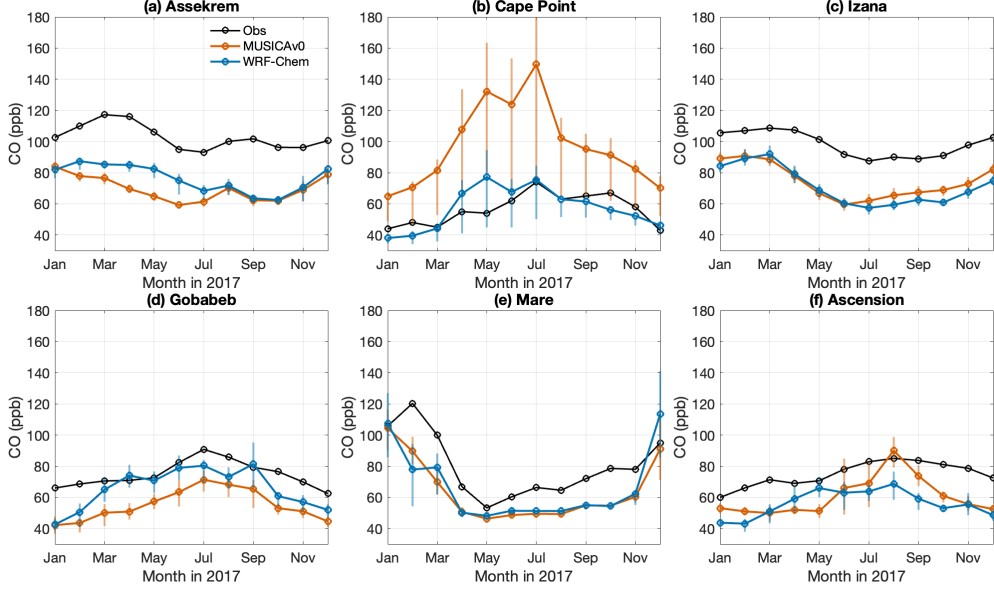

**Figure 6**. Monthly mean CO (ppb) from in situ observations (black), MUSICAv0 (red), and WRF-
Chem (blue) during 2017 at (a) Assekrem, (b) Cape Point, (c) Izana, (d) Gobabeb, (e) Mare and
(f) Ascension (see Figure 1b for locations). Monthly means are calculated from 3-hourly data. The
range for each data point shows the variation of the 3-hourly data on that day (25% quantile to
75% quantile). Observational data are from World Data Centre for Greenhouse Gases (WDCGG).


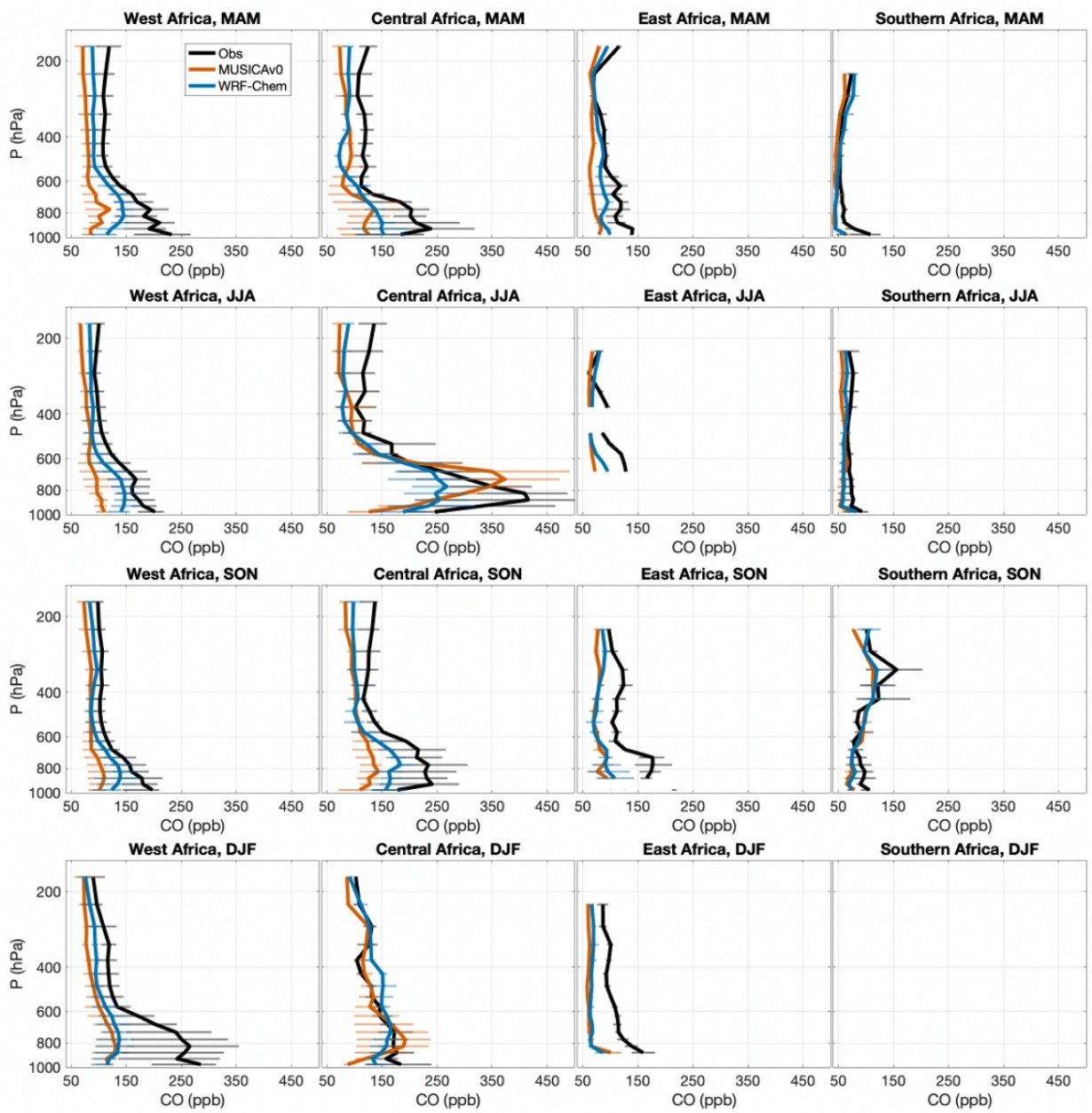

**Figure 7**. Vertical profiles of CO (ppb) from the In-service Aircraft for a Global Observing System
(IAGOS) measurements (black) and corresponding model output from MUSICAv0 (red), and
WRF-Chem (blue) during different seasons in 2017 over West Africa, Central Africa, East Africa,
and Southern Africa. North Africa is not shown due to data availability. Seasonal mean profiles
with the variation of the data in the pressure layer (25% quantile to 75% quantile) in MAM (March,
April, and May), JJA (June, July, and August), SON (September, October, and November), and
DJF (December, January, and February) are shown.

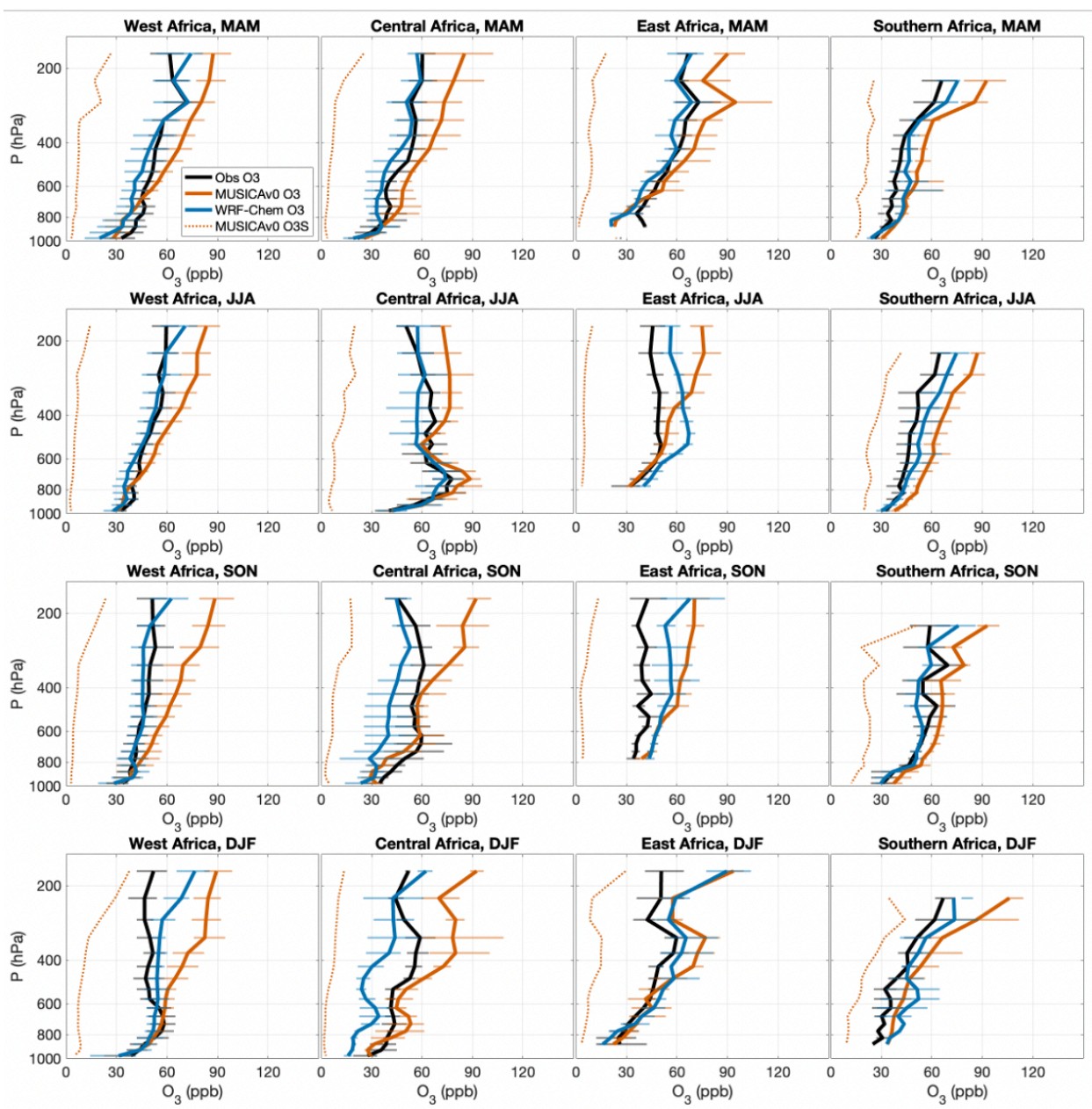

**Figure 8**. Vertical profiles of O₃ (ppb) from the In-service Aircraft for a Global Observing System (IAGOS) measurements (black) and corresponding model output from MUSICAv0 (red), and WRF-Chem (blue) during different seasons in 2017 over West Africa, Central Africa, East Africa, and Southern Africa. North Africa is not shown due to data availability Seasonal mean profiles with the variation of the data in the pressure layer (25% quantile to 75% quantile) in MAM (March, April, and May), JJA (June, July, and August), SON (September, October, and November), and DJF (December, January, and February) are shown. The dash red lines represent O3S (stratospheric ozone tracer) from the MUSICAv0 simulation.

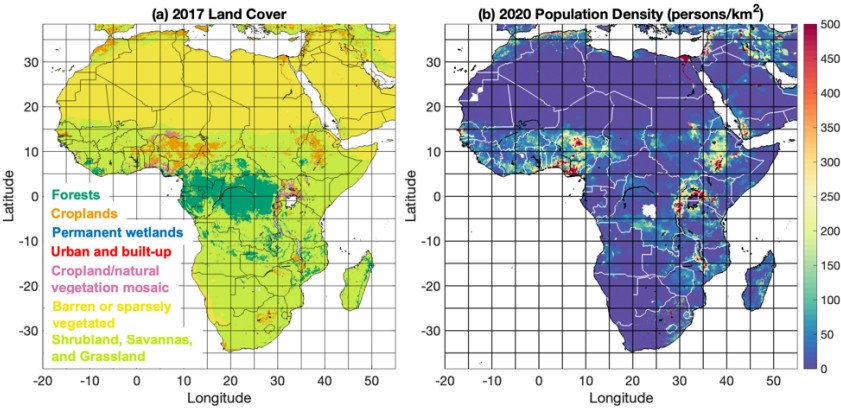

**Figure 9**. (a) Land cover in 2017 and (b) population density (persons/km²) in 2020 over Africa.
Land cover data is from MODIS/Terra+Aqua Land Cover Type Yearly L3 Global product
(resolution: 0.05 degree) (Friedl et al., 2022). Cropland/Natural Vegetation Mosaics means
Mosaics of small-scale cultivation (40-60%) with natural tree, shrub, or herbaceous vegetation.
Population density data is from the Gridded Population of the World, Version 4 (GPWv4),
Revision 11 (CIESIN, 2018).

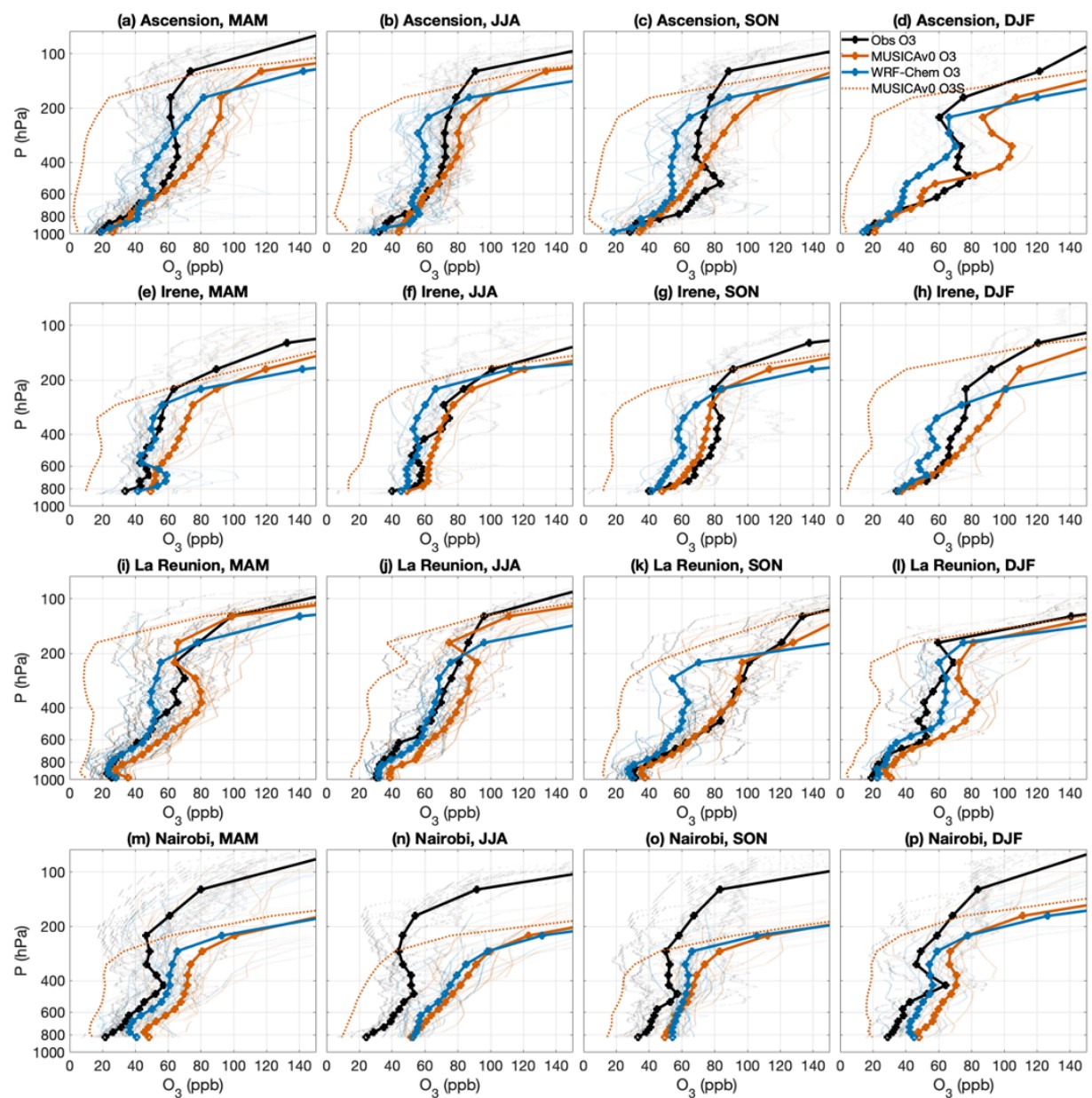


**Figure 10**. Vertical profiles of O₃ (ppb) from Ozonesondes (black) and corresponding model
output from MUSICAv0 (red), and WRF-Chem (blue) for each season of 2017. The thick lines
denote the seasonal mean profiles and the thin lines denote the individual profiles. The dash red
lines represent O3S (stratospheric ozone tracer) from the MUSICAv0 simulation. Ozonesonde data
at Ascension in (a) MAM (March, April, and May), (b) JJA (June, July, and August), (c) SON
(September, October, and November), and (d) DJF (December, January, and February) are shown.
(e-h), (i-l), and (m-p) are the same as (a-d), except for Irene, La Reunion, and Nairobi, respectively.
Locations of the sites are shown in Figure 1b.


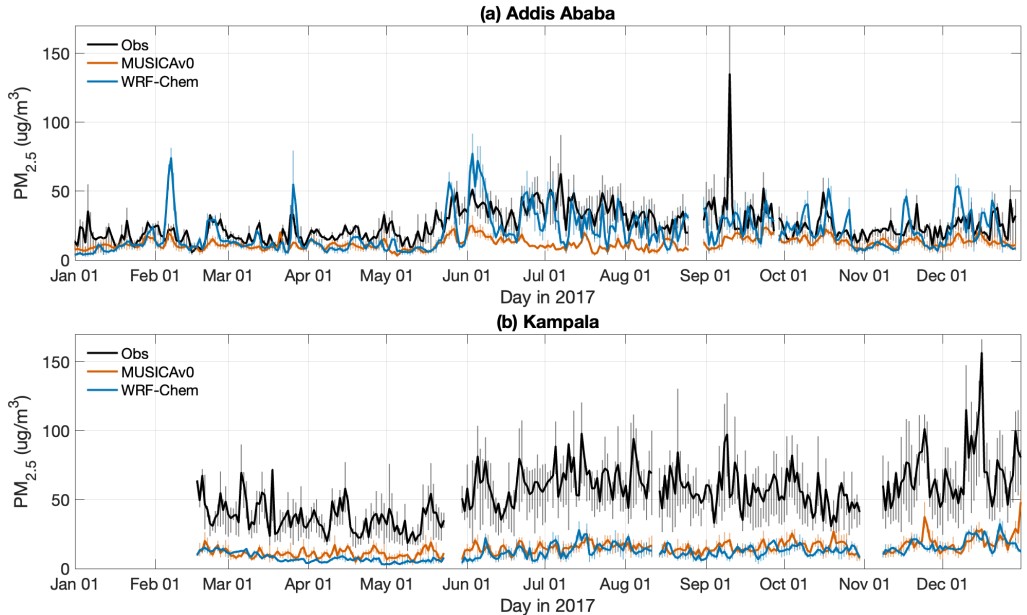

Figure 11. Daily mean PM$_{2.5}$ from in situ observations (black), MUSICAv0 (red), and WRF-Chem (blue) during 2017 at (a) Addis Ababa and (b) Kampala. Daily means are calculated from 3-hourly data. The shown range for each data point shows the variation on that day (25% quantile to 75% quantile). Locations of the sites are shown in Figure 1b.

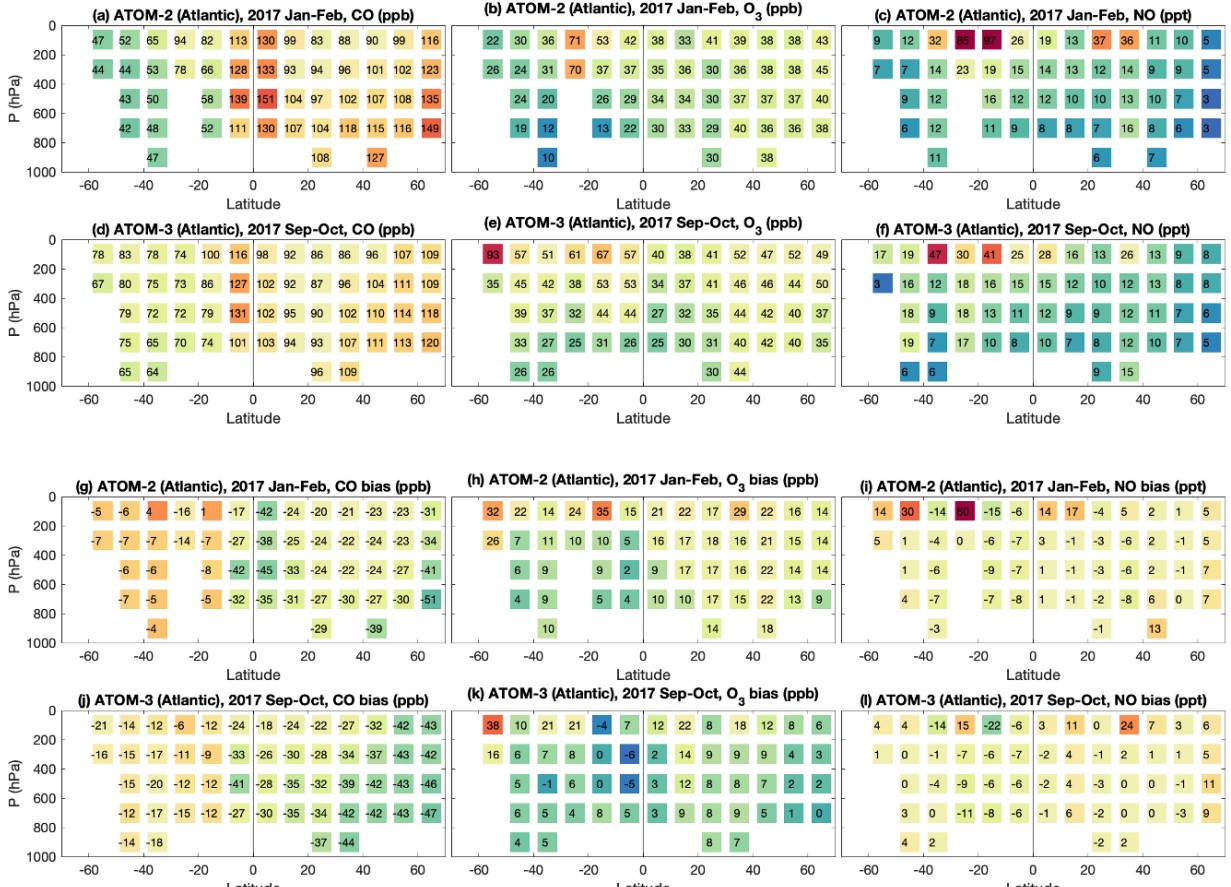

1301
**Figure 12**. Observations of (a) CO (ppb), (b) O₃ (ppb), and (c) NO (ppt) over Atlantic Ocean
during ATom-2 and ATom-3 (d-f). (g-l) corresponding model biases against ATOM observations.
The ATom airborne measurements and corresponding MUSICAv0 model results are binned to 10-
degree latitude and 200-hPa pressure bins. The values of mean biases for each latitude and pressure
bin are labeled in the figure.

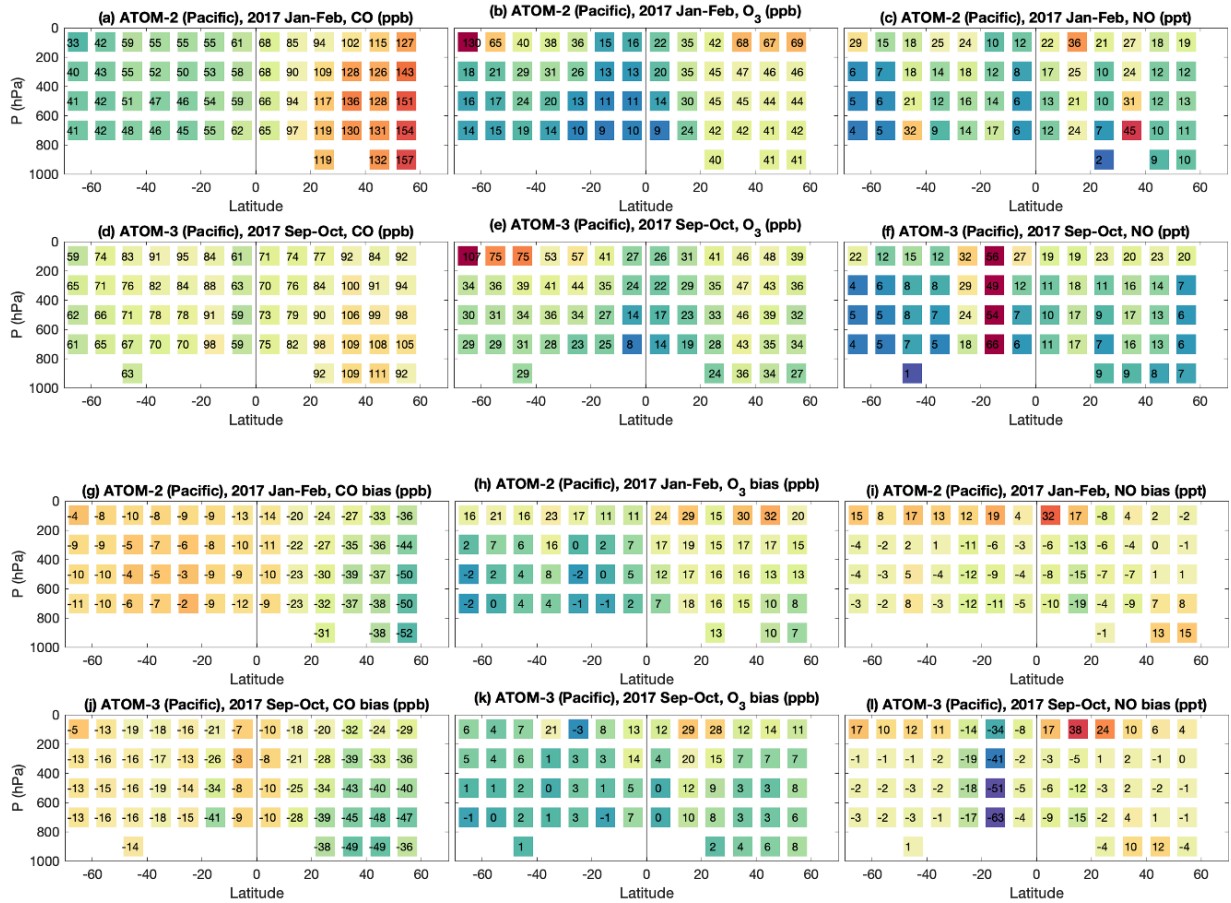

**Figure 13**. Same as Figure 9 but for over the Pacific Ocean.

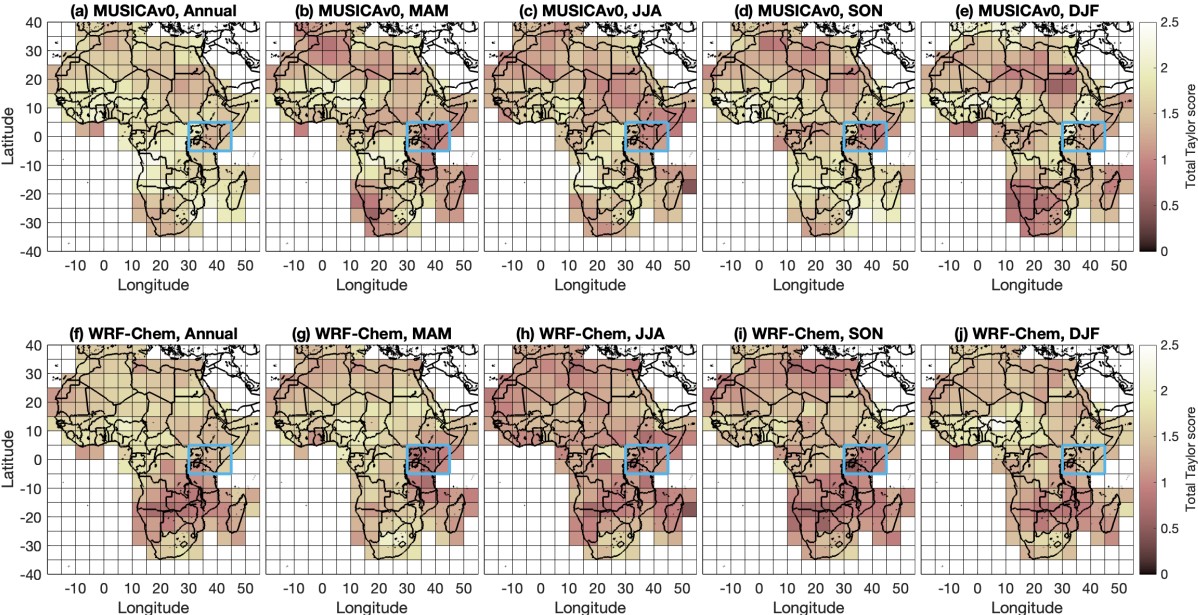

**Figure 14**. Spatial distribution of total Taylor score of MUSICAv0 and (f-j) WRF-Chem compared
to satellite retrievals. In each 5° × 5° (latitude × longitude) pixel, Taylor scores of the model
compared to three satellite products (e.g., MOPITT CO column retrievals, OMI tropospheric $NO_2$
column retrievals, and MODIS AOD) are calculated separately (as shown in Figure S12). Taylor
score against each satellite product ranges from 0 to 1. And then three Taylor scores are summed
up to obtain the shown total Taylor score (ranges from 0 to 3). Total Taylor score of MUSICAv0
for (a) 2017, (b) MAM (March, April, and May), (c) JJA (June, July, and August), (d) SON
(September, October, and November), and (e) DJF (December, January, and February) are shown.
The blue box highlights a potential region for future field campaigns and/or in situ observations.
(f-j) are similar to (a-e) except for WRF-Chem.

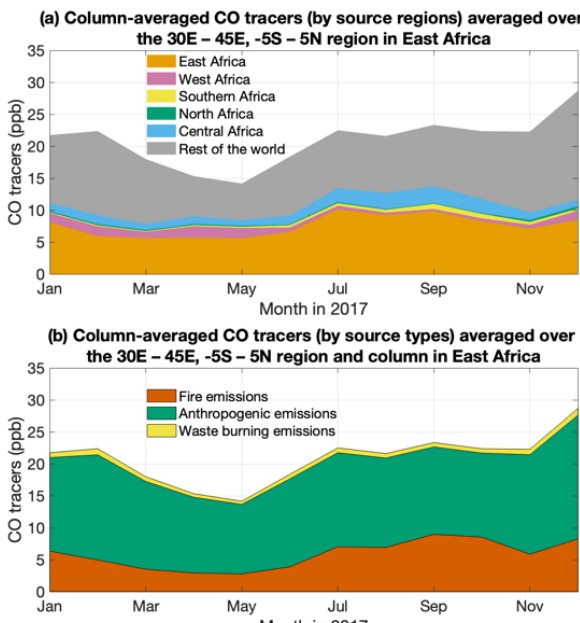

**Figure 15**. Monthly time series of column-averaged CO tracers in the 30°E – 45°E, -5°S – 5°N
region in East Africa. (a) CO tracers of emissions from North Africa (green), West Africa (pink),
East Africa (orange), Central Africa (blue), Southern Africa (yellow), and the rest of the world
(grey). (b) CO tracers of fire emissions (red), anthropogenic emissions (green), and waste burning
emissions (yellow).