# Peer review of "Application of the Multi-Scale Infrastructure for Chemistry and Aerosols version 0 (MUSICAv0) for air quality research in Africa"

_Geoscientific Model Development, 2023_

## Referee Comment (RC1)

The paper "Application of the Multi-Scale Infrastructure for Chemistry and aerosol version 0 (MUSICAv0) for air quality in Africa by Tang et.al

Suggest change in the title to show with the application is research or forecasting of air quality?

The paper describes the new model and compares it with currently existing models in use by the community and satellite measurements and makes a recommendation as the most appropriate tool for use in East Africa.

The development of this model and the intended application is very timely as Africa is experiencing an unprecedented economic and population growth. Of all of the developing world, Africa has experienced the highest urban growth rate during the last two decades at 3.5% per year, and this rate is expected to hold into 2050. With increased urbanization rates, Africa is facing increasingly serious urban air quality problems.

Despite the seriousness of the impact of air pollution in the continent studies addressing this problem are very few. While the paper indicates the scarcity of ground-based observations, the many environmental challenges, the core reason for these is not addressed. The colonial legacy and the continued neo colonialism will continue to hinder the efforts of environmental challenges. While the scientific research helps show the problems, will not contribute to the deeper crisi and problems the continent faces as it turns out to be the ground for scramble for economic power houses in North America, China, and Europe.

While this is a purely scientific paper, the introduction that addresses the environmental challenges, poverty (A continent with vast natural resources suffering of poverty), and deficit of knowledge needs to  at least mention the political forces that caused the current situation in Africa.

Line 73-74 "Atmospheric chemistry modeling is a useful tool to perform research on air quality conditions and evolution".. doesn't make sense. The model can be used to provide air quality forecasts or perhaps understand the chemical processes.

Line 100-103; "MUSICAv0 reproduces the results of WRF-Chem" is not enough justification to introduce a new model. The unique and added values of MUSICAv0 should be clearly stated. Reproducing what other models can do is not enough justification.

Line 264: MODIS AOD

Satellite-based aerosol retrievals of aerosol optical depth (AOD) provide exceptional spatial coverage but suffer over bright surfaces and regions with complicated surface terrain. East Africa has a complicated terrain due to mountains and the rift valley. This issue should be addressed when comparing MODIS AOD products and compare with MUSICAv0) model performance.

Line 274-275:" CO is a good tracer of anthropogenic and biomass burning emissions and modeled CO tracers are used."

The tracer method has been widely used to identify certain emission sources. For biomass burning, acetonitrile (de Gouw et al.,2003, G. Wang, et al.,2016 hydrogen cyanide  Hornbrook et al.2011), methyl chloride (de Gouw et al., 2004 furans (Coggon et al., 1016 ), levoglucosan (Bhattarai et al., 2019),

The authors should provide justification for using just CO.

**Section 3**

Considering the flow of pollutants across the different regions in Africa, across all regions, local emissions from biomass burring and anthropogenic sources impact ambient $PM_{2.5}$ the most within the source region itself.

The impacts of emissions from one region on the annual-average ambient $PM_{2.5}$ in another region depends strongly on meteorology. For example, the combustion sources in East Africa contribute 20% of the annual-average ambient $PM_{2.5}$ over Central Africa. (see https://agupubs.onlinelibrary.wiley.com/doi/full/10.1029/2022GH000673

The authors should address the impact of meteorology in distributions of pollutants across the regions.

Line 372 "Column peak in September likely driven by fire emissions" – This is incorrect.

In the southern, western, and eastern regions of Africa, the trees and grass of the savannah biome become extremely flammable during the dry season, which lasts from May to October in southern Africa and January to April in west and east Africa. September is just the end of the rainy season in East Africa

Line 389: "There are two surface PM2.5 sites in East Africa" In addition to the Embassy Sites, MAIA project has several instruments in Addis Ababa.

The authors need to mention the type of the instruments at the sites etc.

Line 415-416: "Overestimation of CO by MUSICAv0 should be due to overestimation of anthropogenic emissions from Southern Africa"

This depends on the period. Southern Africa Fire season goes from *April to August*.

Line 493-494- "And then three…." Starting a sentience with And is odd

Please consider revising paragraph starting line 508-520

For example, Line 514-515, "Anthropogenic emissions play a more important role in the region compared to East Africa "The paragraph starts addressing East Africa, what is the region mentioned in the sentence above that is being compared with east Africa?

Line 517-518 "Note that the source analysis using model tracers may be subject to uncertainties in the emission inventories"

Needs to be clarified which inventories are considered? It should also be pointed out that Most modeling studies have relied on emission factors measured outside the continent of Africa (Lamarque et al. 2010; Klimont, Smith, and Cofala 2013; Pokhrel et al. 2021).

---

## Author Comment (AC1)

[revised manuscript text omitted]

**Page 14: [1] Deleted                Wenfu Tang              6/17/23 8:23:00 PM**

---

## Author Comment (AC2)

**Reviewer 1**

1. The paper "Application of the Multi-Scale Infrastructure for Chemistry and aerosol version 0 (MUSICAv0) for air quality in Africa by Tang et.al. Suggest change in the title to show with the application is research or forecasting of air quality?
**Response**: We changed the title to "Application of the Multi-Scale Infrastructure for Chemistry and Aerosols version 0 (MUSICAv0) for air quality research in Africa".

2. The paper describes the new model and compares it with currently existing models in use by the community and satellite measurements and makes a recommendation as the most appropriate tool for use in East Africa. The development of this model and the intended application is very timely as Africa is experiencing an unprecedented economic and population growth.
**Response**: Thank you!

3. Of all of the developing world, Africa has experienced the highest urban growth rate during the last two decades at 3.5% per year, and this rate is expected to hold into 2050. With increased urbanization rates, Africa is facing increasingly serious urban air quality problems. Despite the seriousness of the impact of air pollution in the continent studies addressing this problem are very few. While the paper indicates the scarcity of ground-based observations, the many environmental challenges, the core reason for these is not addressed. The colonial legacy and the continued neo colonialism will continue to hinder the efforts of environmental challenges. While the scientific research helps show the problems, will not contribute to the deeper crisi and problems the continent faces as it turns out to be the ground for scramble for economic power houses in North America, China, and Europe. While this is a purely scientific paper, the introduction that addresses the environmental challenges, poverty (A continent with vast natural resources suffering of poverty), and deficit of knowledge needs to at least mention the political forces that caused the current situation in Africa.
**Response**: We added the following statement to Introduction:
"Intergovernmental Panel on Climate Change (IPCC) computes a human vulnerability metric from existing challenges such as poverty, access to health care plus expected mortality for climate hazards such as heat, drought, flood, fires and constraints to adaptation like funding, and government infrastructure (Moss et al., 2001). Many regions in Africa exhibit the most extreme values for this metric."
This statement hints at political instability. Further discussion on political forces beyond this is beyond the scope of this paper.

4. Line 73-74 "Atmospheric chemistry modeling is a useful tool to perform research on air quality conditions and evolution".. doesn't make sense. The model can be used to provide air quality forecasts or perhaps understand the chemical processes.
**Response**: We revised the sentence to "Atmospheric chemistry modeling is a useful tool to provide air quality forecasts and to understand chemical processes."

5. Line 100-103; "MUSICAv0 reproduces the results of WRF-Chem" is not enough justification to introduce a new model. The unique and added values of MUSICAv0 should be clearly stated. Reproducing what other models can do is not enough justification.

**Response**: Lines 100-103 only explains the reason for including WRF-Chem in this study. We compared MUSICAv0 with WRF-Chem because MUSICAv0 with Africa refinement is newly developed while WRF-Chem has been previously used for African air quality. Reproducing the results of WRF-Chem is only one of the conclusions of this paper. We evaluated MUSICAv0 against in situ observations and compared MUSICAv0 results with satellite retrievals to justify the usage of this new model. The uniqueness and added values of MUSICAv0 is stated in the Introduction and Section 2.1.

6. Line 264: MODIS AOD. Satellite-based aerosol retrievals of aerosol optical depth (AOD) provide exceptional spatial coverage but suffer over bright surfaces and regions with complicated surface terrain. East Africa has a complicated terrain due to mountains and the rift valley. This issue should be addressed when comparing MODIS AOD products and compare with MUSICAv0 model performance.

**Response**: We used Deep Blue Aerosol retrievals in this study to address the issues with bright surfaces. Deep blue uses maps and libraries of surface reflectance in the blue channels to account for the surface signal as well as spectral reflectance ratios. This method works best over bright land surfaces but can also retrieve aerosols over most vegetated targets. We thank the reviewer for pointing out the issue of complex terrain due to mountains and the rift valley. We have added the following statement in Section 2.11:

"East and Southern Africa have complex terrain due to mountains and rift valleys. This may lead to some uncertainties in MODIS AOD retrievals."

7. Line 274-275:" CO is a good tracer of anthropogenic and biomass burning emissions and modeled CO tracers are used." The tracer method has been widely used to identify certain emission sources. For biomass burning, acetonitrile (de Gouw et al.,2003, G. Wang, et al.,2016 hydrogen cyanide Hornbrook et al.2011), methyl chloride (de Gouw et al., 2004 furans (Coggon et al., 1016), levoglucosan (Bhattarai et al., 2019),   The authors should provide justification for using just CO.

**Response**: CO and the tracers mentioned by the reviewer are all valid tracers. However, CO is a commonly used tracer in models. Acetonitrile, methyl chloride, and levoglucosan are used as tracers in observation-based studies. We added the following statement to Section 3 for explanation.

"CO is a commonly used tracer in models with only one photochemical sink and an intermediate lifetime (e.g., Tang et al., 2019). CO tracers also allow clear identification of simulated anthropogenic and biomass burning contributions. Therefore, tagging CO is computationally efficient and tagged CO is relatively reliable as a tracer in models."

8. Considering the flow of pollutants across the different regions in Africa, across all regions, local emissions from biomass burring and anthropogenic sources impact ambient PM2.5 the most within the source region itself.  The impacts of emissions from one region on the annual-average ambient PM2.5 in another region depends strongly on meteorology. For example, the combustion sources in East Africa contribute 20% of the annual-average ambient PM2.5 over Central Africa. (see https://agupubs.onlinelibrary.wiley.com/doi/full/10.1029/2022GH000673 The authors should address the impact of meteorology in distributions of pollutants across the regions.

**Response**: We agree with the reviewer that meteorology has a significant impact on the distributions of pollutants across the regions. The modeled CO tracers used in this study are subject

to model meteorology and the results have taken into consideration of meteorology and transport. In Figure 4, for a specific region, the contribution of CO tracers from different regions is a result of both emissions and meteorology (transport). We added the following statement to Section 3 to explain that we accounted meteorology:

"Meteorology has a significant impact on the distributions of pollutants across the regions (e.g., Gordon et al., 2023). The CO tracers in the model go through the same model processes (e.g., transport) as CO. Therefore, the source contribution shown by the CO tracers is a result of both emissions and transport."

9. Line 372 "Column peak in September likely driven by fire emissions" – This is incorrect. In the southern, western, and eastern regions of Africa, the trees and grass of the savannah biome become extremely flammable during the dry season, which lasts from May to October in southern Africa and January to April in west and east Africa. September is just the end of the rainy season in East Africa.

**Response**: Fire emissions from central Africa peaks in September and can be transported to East Africa. In addition, the definition of East Africa encompasses a large area of the southern hemisphere, which is subject to fires at this time. Figure 2 shows this for CO. To address the reviewer's comment, we modified the statement. See below for the updated statement.

"The biases in CO column and tropospheric $NO_2$ column peak in September. One possible driver could be fire emissions from other regions (Figure 4), however, further studies will be needed to address this."

10. Line 389: "There are two surface PM2.5 sites in East Africa" In addition to the Embassy Sites, MAIA project has several instruments in Addis Ababa. The authors need to mention the type of the instruments at the sites etc.

**Response**: The instrument types and other relevant information are described in Section 2.7.

"At the U.S. embassies, regulatory-grade monitoring data are collected with Beta Attenuation Monitors (BAMs), using a federal equivalent monitoring method, with an accuracy within 10% of federal reference methods (Watson et al., 1998; U.S. EPA, 2016). These instruments are operated by the U.S. State Department and the U.S. EPA, and data are available through AirNow (https://www.airnow.gov/international/us-embassies-and-consulates/)."

We only used publicly available measurements for the year 2017. We were not able to find measurements from MAIA project for 2017 while conducting this study. Nevertheless, we changed the statement "There are two surface PM2.5 sites in East Africa" to "We compare the model results with PM2.5 measurements from two surface sites in East Africa" to avoid any confusion.

11. Line 415-416: "Overestimation of CO by MUSICAv0 should be due to overestimation of anthropogenic emissions from Southern Africa" This depends on the period. Southern Africa Fire season goes from April to August.

**Response**: The conclusion "Overestimation of CO by MUSICAv0 should be due to overestimation of anthropogenic emissions from Southern Africa" is based on source contribution by model CO tracers for Cape Point in Figures S4 and S5 (also shown below). As shown by the CO tracers, the CO concentration at Cape Point is dominant by anthropogenic emissions from Southern Africa. Fire emissions only contribute to less than 10 ppb (Figure S5b) and cannot be the reason for the overestimation of 40 ppb.

[Figure]

12. Line 493-494- "And then three…." Starting a sentience with And is odd
**Response**: Thank you. We deleted "And".

13. Please consider revising paragraph starting line 508-520
For example, Line 514-515, "Anthropogenic emissions play a more important role in the region compared to East Africa "The paragraph starts addressing East Africa, what is the region mentioned in the sentence above that is being compared with east Africa?
**Response**: This paragraph focus on the 30°E – 45°E, 5°S – 5°N region, which is a sub-region in East Africa rather than the whole East Africa. This paragraph compares this sub-region with East Africa in general. To avoid any confusion, we changed the first sentence of the paragraph to:
"The 30°E – 45°E, 5°S – 5°N region (a sub-region in East Africa) is potentially a favorable location for future field campaign(s)". We also changed "Anthropogenic emissions play a more important role in the region compared to East Africa in general" to "Anthropogenic emissions play a more important role in the 30°E – 45°E, 5°S – 5°N region compared to East Africa in general".

14. Line 517-518 "Note that the source analysis using model tracers may be subject to uncertainties in the emission inventories". Needs to be clarified which inventories are considered? It should also be pointed out that Most modeling studies have relied on emission factors measured outside the continent of Africa (Lamarque et al. 2010; Klimont, Smith, and Cofala 2013; Pokhrel et al. 2021).
**Response**: Thank you for pointing this out. We listed the inventories and added discussion on emission factors:
"Note that the source analyses using model tracers may be subject to uncertainties in the emission inventories, in this case CAMSv5.1, QFED, and the waste burning inventory used here. As discussed above (e.g., Section 3.4), there might be missing sources in the region. In addition, emission factors used in many emission inventories are based on measurements outside the continent of Africa (e.g., Lamarque et al. 2010; Klimont et al., 2013; Pokhrel et al. 2021). It is not clear so far if these emission factors are applicable to emissions in Africa (e.g., Keita et al., 2018). Therefore, a field campaign in the region can help address these issues."

---

## Author Comment (AC3)

**Reviewer 2**

This manuscript provides an assessment of the performance of a new modelling platform applied across Africa. The study is well-aligned to GMD and provides new and interesting results for a domain that is often overlooked. I suggest that revisions are needed before the manuscript could be accepted.

1. I have provided comments on the pdf of the main document and supplementary material in the attached zip file. The main issue that I have is that the analysis did not use many available measurements that could have assisted with an improved analysis of the model and that there is still a lack of discussion of the results with local literature on the key issues that are discussed in the paper. I unpack both of these below and in the attached. I strongly recommend that these are addressed before the manuscript can be published.

**Response**: Thank you. We have addressed the specific comments below and in the attached. Please see below for details, especially the response to Reviewer 2 Comment #2 for updates on measurements used in this study. We included more local literature in the revised manuscript. A full list of newly added references is included below.

Amegah, A.K. and Agyei-Mensah, S., 2017. Urban air pollution in Sub-Saharan Africa: Time for action. Environmental Pollution, 220, pp.738-743.

[revised manuscript text omitted]

Sserunjogi, R., Ssematimba, J., Okure, D., Ogenrwot, D., Adong, P., Muyama, L., Nsimbe, N., Bbaale, M. and Bainomugisha, E., 2022. Seeing the air in detail: Hyperlocal air quality dataset collected from spatially distributed AirQo network. Data in Brief, 44, p.108512.

Swilling, M., Musango, J. and Wakeford, J., 2016. Developmental states and sustainability transitions: prospects of a just transition in South Africa. Journal of Environmental Policy & Planning, 18(5), pp.650-672.

Tang, W., Pfister, G.G., Kumar, R., Barth, M., Edwards, D.P., Emmons, L.K. and Tilmes, S., 2023. Capturing High-Resolution Air Pollution Features Using the Multi-Scale Infrastructure for Chemistry and Aerosols Version 0 (MUSICAv0) Global Modeling System. Journal of Geophysical Research: Atmospheres, 128(7), p.e2022JD038345.

Thompson, A. M., Balashov, N. V., Witte, J. C., Coetzee, J. G. R., Thouret, V., and Posny, F.: Tropospheric ozone increases over the southern Africa region: bellwether for rapid growth in Southern Hemisphere pollution?, Atmos. Chem. Phys., 14, 9855–9869, https://doi.org/10.5194/acp-14-9855-2014, 2014.

Tshehla, C. and Wright, C.Y.: 15 years after the National Environmental Management Air Quality Act: Is legislation failing to reduce air pollution in South Africa?. South African Journal of Science, 115(9-10), pp.1-4, 2019.

Yoshioka, M., Mahowald, N.M., Conley, A.J., Collins, W.D., Fillmore, D.W., Zender, C.S. and Coleman, D.B.: Impact of desert dust radiative forcing on Sahel precipitation: Relative importance of dust compared to sea surface temperature variations, vegetation changes, and greenhouse gas warming. Journal of Climate, 20(8), pp.1445-1467, 2007.

Zhang, D., Du, L., Wang, W., Zhu, Q., Bi, J., Scovronick, N., Naidoo, M., Garland, R.M. and Liu, Y. A machine learning model to estimate ambient PM2. 5 concentrations in industrialized highveld region of South Africa. Remote sensing of environment, 266, p.112713, 2021.

2. Ground-based data are scarce, but they are **not** non-existent. This is a major failing of the paper. As it is a Pan-African study, it is understandable that the authors would not need to use all of the measurements from more local studies as some within the same city would (for example) be within similar grid cells. However, it is not acceptable that they used so few. I have noted in the attached possible datasets that can be used and previous literature that can help in interpretations. For example, in Western Africa there are ground-based measurements from DACCIWA that overlap with at least some of 2017. In South Africa, there are freely available regulatory measurements across the country in 2017 from SAAQIS. My suggestions are not exhaustive, but are examples. I would expect that the authors would search for available data that could be used for model validation. In this manuscript it seems that the search was not yet complete nor comprehensive. I recommend that this gap is filled as this impacts upon the conclusions. For example, in southern Africa, the site used is Cape Point. This is a very unique site and not representative of southern Africa. I can understand why it was used as one of the sites (it is a well-characterized GAW site), but I cannot understand why it was the ONLY site used. There are regulatory grade monitors across the country. Using a handful of these could at least provide a more comprehensive assessment of

the performance of the model across more of the region. I do understand that accessing data across Africa can be harder as there is not one central place such as in the US. However, such a manuscript should aim to represent the region it is modelling as well as it can, which includes trying to use as many measurements that are available.

**Response**: We include (1) AERONET AOD, (2) SAAQIS PM$_{2.5}$, CO, NO$_2$, and O$_3$ data, and (3) PM2.5 data at 4 West Africa sites as suggested by the reviewer. See the figures below and text in the revise manuscript. We believe the satellite and in situ data (including the newly added ones) included in the paper are enough to provide an initial look of model performance in Africa. In the future, we will run more recent years, include more observations when available and relevant.

[Figure]

**Figure 1.** Model grid, in situ observations used in this study, and sub-regions in Africa. (a) MUSICAv0 model grid developed for Africa in this study (black), domain boundary of the WRF-Chem simulation compared in this study (shown by green box), observations from the Atmospheric Tomography Mission (ATom) field campaign 2 (ATom-2; 2017 Jan to 2017 Feb; pink) and ATom-3 (2017 Sep to 2017 Oct; yellow). (b) Sub-regions in Africa are shown, namely North Africa (green), West Africa (pink), East Africa (orange), Central Africa (blue), and Southern Africa (yellow). Location of in situ observations are labeled on the map. Flight tracks of the In-service Aircraft for a Global Observing System (IAGOS) are shown with black lines. Four ozonesonde sites are shown by pentagrams (Ascension, Irene, Nairobi, and La Reunion); six sites from the World Data Centre for Greenhouse Gases are shown by triangles (Assekrem, Cape Point, Izana, Gobabeb, Mare, and Ascension); surface sites for PM$_{2.5}$ are shown by squares (Addis Ababa and Kampala in East Africa; Abidjan and Cotonou in West Africa); AErosol RObotic NETwork (AERONET) sites are shown with diamond; South Africa Air Quality Information System (SAAQIS) sites are shown with blue circles.

[Figure]

**Figure 4**. Comparisons of MUSICAv0 and WRF-Chem simulations and MODIS and AERONET AOD at 550 nm in 2017. (a-d) Averaged MODIS and AERONET AOD in MAM (March, April, and May), JJA (June, July, and August), SON (September, October, and November), and DJF (December, January, and February). (e-h) MUSICAv0 model biases against MODIS and AERONET AOD in MAM, JJA, SON, and DJF. (i-l) is the same as (e-h) but for WRF-Chem. All data are gridded to 0.25 degree × 0.25 degree for plotting. AERONET AOD in (a-d) and model bias against AERONET AOD in (e-l) are shown by the circles overlayed on the map.

[Figure]

**Figure S7**. Weekly mean PM$_{2.5}$ from in situ observations (black), MUSICAv0 (red), and WRF-Chem (blue) during 2017 at Abidjan site representing domestic fires (ADF) emissions, Abidjan site representing waste burning at landfill (AL), Abidjan site representing traffic (AT), and Cotonou site representing traffic (CT). The three sites in Abidjan are close in space.

[Figure]

**Figure S8**. Comparisons of MUSICAv0 and WRF-Chem simulations and with SAAQIS CO measurements in 2017. (a-d) Averaged SAAQIS CO measurements in MAM (March, April, and May), JJA (June, July, and August), SON (September, October, and November), and DJF (December, January, and February). (e-h) MUSICAv0 model biases against SAAQIS CO measurements in MAM, JJA, SON, and DJF. (i-l) is the same as (e-h) but for WRF-Chem.

[Figure]

**Figure S9**. Same as Figure S8 but for $NO_2$.

[Figure]

**Figure S10**. Same as Figure S8 but for $O_3$.

[Figure]

**Figure S11**. Same as Figure S8 but for PM$_{2.5}$.

3. In addition, there are gaps in the referenced literature that could support some of the analyses and provide more context for these (e.g. are these trends seen in other papers for some key aspects). Looking through, it seems that global analyses and references are used more than local studies. For example, the bias in Kampala could be because of the complex spatial distribution of PM2.5. The are two papers from Makerere University noted in attached that do not have data in 2017, but can support the spatial variability of PM2.5.

Also, for example, for the timing of fire events and impacts on pollution, there is a lot of literature across the regions. A starting point are papers such as,

- Archibald, S.: Managing the human component of fire regimes: lessons from Africa, Philos. T. R. Soc. B., 371, 20150346, https://doi.org/10.1098/Rstb.2015.0346, 2016.
- Archibald, S., Scholes, R. J., Roy, D. P., Roberts, G., and Boschetti, L.: Southern African fire regimes as revealed by remote sensing, Int. J. Wildland Fire, 19, 861–878, https://doi.org/10.1071/WF10008, 2010.

**Response**: We added more references of local studies. See response to Reviewer 2 Comment #1 for a full list of newly added references. Below are references we added specifically for fires in Southern Africa and PM2.5 in East Africa:

Archibald, S., Roy, D.P., van Wilgen, B.W. and Scholes, R.J.: What limits fire? An examination of drivers of burnt area in Southern Africa. Global Change Biology, 15(3), pp.613-630, 2009.

Archibald, S., Scholes, R.J., Roy, D.P., Roberts, G. and Boschetti, L., 2010. Southern African fire regimes as revealed by remote sensing. International Journal of Wildland Fire, 19(7), pp.861-878.

Archibald, S., 2016. Managing the human component of fire regimes: lessons from Africa. Philosophical Transactions of the Royal Society B: Biological Sciences, 371(1696), p.20150346.

Atuhaire, C., Gidudu, A., Bainomugisha, E. and Mazimwe, A., 2022. Determination of Satellite-Derived PM2. 5 for Kampala District, Uganda. Geomatics, 2(1), pp.125-143.

Kalisa, E., Nagato, E.G., Bizuru, E., Lee, K.C., Tang, N., Pointing, S.B., Hayakawa, K., Archer, S.D. and Lacap-Bugler, D.C.: Characterization and risk assessment of atmospheric PM2. 5 and PM10 particulate-bound PAHs and NPAHs in Rwanda, Central-East Africa. Environmental science & technology, 52(21), pp.12179-12187, 2018.

Kinney, P.L., Gichuru, M.G., Volavka-Close, N., Ngo, N., Ndiba, P.K., Law, A., Gachanja, A., Gaita, S.M., Chillrud, S.N. and Sclar, E., 2011. Traffic impacts on PM2. 5 air quality in Nairobi, Kenya. Environmental science & policy, 14(4), pp.369-378.

Okure, D., Ssematimba, J., Sserunjogi, R., Gracia, N.L., Soppelsa, M.E. and Bainomugisha, E., 2022. Characterization of ambient air quality in selected urban areas in uganda using low-cost sensing and measurement technologies. Environmental Science & Technology, 56(6), pp.3324-3339.

4. Another example, ozone exchange between stratosphere and troposphere is an important component to the paper – but what is known about it in the domain (or even some regions)? This paper focuses on just one year and a few data sources, so other studies on this could help to support and provide context for this study.
**Response**: We agree with the reviewer that ozone exchange between stratosphere and troposphere over Africa is an important topic to study and we hope there will be more follow-up studies in the future. It is also true with one year of model simulation and data in this paper we cannot fully address this question. We added more relevant references:

Leclair De Bellevue, J., Réchou, A., Baray, J. L., Ancellet, G., and Diab, R. D.:Signatures of stratosphere to troposphere transport near deep convective events in the southern subtropics, J. Geophys. Res., 111, D24107, doi:10.1029/2005JD006947,2006.

Mkololo, T., Mbatha, N., Sivakumar, V., Bègue, N., Coetzee, G. and Labuschagne, C.: Stratosphere–Troposphere exchange and O3 variability in the lower stratosphere and upper troposphere over the irene SHADOZ site, South Africa. Atmosphere, 11(6), p.586, 2020.

Oluleye, A. and Okogbue, E.C.: Analysis of temporal and spatial variability of total column ozone over West Africa using daily TOMS measurements. Atmospheric Pollution Research, 4(4), pp.387-397, 2013.

Clain, G., Baray, J. L., Delmas, R., Diab, R., Leclair de Bellevue, J., Keckhut, P., Posny, F., Metzger, J. M., and Cammas, J. P.: Tropospheric ozone climatology at two Southern Hemisphere tropical/subtropical sites, (Reunion Island and Irene, South Africa) from ozonesondes, LIDAR, and in situ aircraft measurements, Atmos. Chem. Phys., 9, 1723–1734, https://doi.org/10.5194/acp-9-1723-2009, 2009.

5. I have tried to highlight some areas that I am familiar with where other studies can assist, however, I don't know all the literature related to this topic. So I recommend that for the key points, that other literature in these regions is looked for to provide such context and support of the findings. I have provided more detailed comments on the attached pdfs.

**Response**: Thank you. We have included more references to the manuscript for the key points. A full list of newly added references is included in the response to Reviewer 2 Comment #1. The detailed comments in the attached pdf have been copied to this document and addressed below.

6. Lines 65-67: I understand that only a selection is needed, but I recommend including some more references. There are many studies on air quality and none of those listed seem to be from African-based researchers.

**Response**: We added the following references for "degraded air quality is an example of a severe environmental challenge with growing importance in Africa":

Kinney, P.L., Gichuru, M.G., Volavka-Close, N., Ngo, N., Ndiba, P.K., Law, A., Gachanja, A., Gaita, S.M., Chillrud, S.N. and Sclar, E., 2011. Traffic impacts on PM2. 5 air quality in Nairobi, Kenya. Environmental science & policy, 14(4), pp.369-378.

Okure, D., Ssematimba, J., Sserunjogi, R., Gracia, N.L., Soppelsa, M.E. and Bainomugisha, E., 2022. Characterization of ambient air quality in selected urban areas in uganda using low-cost sensing and measurement technologies. Environmental Science & Technology, 56(6), pp.3324-3339.

Naiker, Y., Diab, R.D., Zunckel, M. and Hayes, E.T., 2012. Introduction of local Air Quality Management in South Africa: overview and challenges. Environmental science & policy, 17, pp.62-71.

Amegah, A.K. and Agyei-Mensah, S., 2017. Urban air pollution in Sub-Saharan Africa: Time for action. Environmental Pollution, 220, pp.738-743.

7. Lines 90-91: "This is critical because some of the environmental issues in Africa are coupled" As worded it seems this is only true in Africa, but this is a more general statement. I recommend rewording.

**Response**: We changed "some of the environmental issues in Africa are coupled" to "some of the environmental issues are coupled".

8. The contribution by dust is important across most regions, and it is noted often that the model(s) are struggling to perform well in areas impacted by dust. However, how dust was modelled in MUSICA and WRF-Chem is not clear below. I recommend it is added.

**Response**: We added the following description of dust in MUSICAv0 to Section 2.1:
"The generation of desert dust particles in MUSICAv0 is calculated based on the Dust Entrainment and Deposition Model (Mahowald et al., 2006; Yoshioka et al., 2017). Dust emissions calculation is sensitive to the model surface wind speed. The dust aerosol processes in the MUSICAv0 simulation are simulated based on the MAM4 model (Liu et al., 2016). MAM4 has 4 modes –

Aitken, accumulation coarse, and primary carbon modes. Dust is mostly in the accumulation and coarse modes."

We also added the following description of dust in WRF-Chem to Section 2.2:
"The dust aerosol processes in the WRF-Chem simulation are simulated based on the Goddard Global Ozone Chemistry Aerosol Radiation and Transport (GOCART) model (Chin et al., 2002). Specifically, the dust emission scheme is following the GOCART emission treatment (Ginoux et al., 2001), which is a function of 10-m wind speed, soil moisture, and soil erosion capability. The atmospheric processes of dust are simulated based on the mass mixing ratio and size distribution that has been divided into 5 size bins with effective radii of 0.73, 1.4, 2.4, 4.5 and 8.0 μm. The dust dry and wet depositions are also treated following the GOCART scheme (Chin et al., 2002)."

9. Lines 134-135: "We use the Copernicus Atmosphere Monitoring Service Global Anthropogenic emissions, (CAMS-GLOB-ANT) version 5.1 (Soulie et al., 2023) for anthropogenic emissions …".
 I recommend some explanation of how this emission inventory was selected. There is also DACCIWA that has been developed specifically for Africa. Was that emission inventory considered? I know there are many considerations in selecting an emission inventory, so I am not suggesting a change in the method. However, I do recommend that the reasons for this choice is described.
**Response**: We added the following statement to Section 2.1:
"CAMS-GLOB-ANT version 5.1 (Soulie et al., 2023) is one of the most widely used global inventories for anthropogenic emissions. CAMS-GLOB-ANT version 5.1 has been implemented in MUSICAv0, and evaluated in our previous studies (Tang et al., 2022, 2023; Jo et al., 2023). CAMS-GLOB-ANT version 5.1 does not include information from DACCIWA, however, a future version of CAMS-GLOB-ANT is expected to include DACCIWA for Africa. In future work on this topic, we plan to make use of regional emissions inventories, such as  the DACCIWA emission inventory."

10. Section 2.3: For Atom and IAGOS measurements it would be helpful to see how many measurements points were compared and for each season. Below it is noted that N Africa IAGOS data was not used due to data availability issues - but then what threshold was used? Understanding the limitations in these short-term data (i.e. specific flights) will help readers to better understand the comparison to the model outputs (e.g. especially if the measurement data were few).  A simple table could be added to the supp material to show points per measurements per season per region.
**Response**: As we use 2-minute merged ATom measurements, there are 2796 data points in ATom-2 (January–February 2017) and 3369 data points in ATom-3 (September–October 2017). We added this information to Section 2.3 where we describe ATom data. North Africa IAGOS data is not used because there are no data in some layers and it is not possible to obtain a profile for North Africa using IAGOS data. Please see below:

[Figure]

11. Section 2.7 Surface PM2.5: Why were only these stations used? There is a regulatory network across South Africa that also has regulatory-grade PM2.5 measurements. Why weren't at least some of these sites included? To me this is an important issue that needs to be addressed.
Also, in this same network there are measurements of CO, NO/NOx/NO2 and ozone. I understand the focus wasn't South Africa only, so all sites don't need to be used. But it is strange that none were used as they are freely available on https://saaqis.environment.gov.za/
and there were sites measuring these parameters in 2017.

**Response**: Thank you for your suggestion. We included SAAQIS data in the study. See to Reviewer 2 Comment #2 for figures. We added a description of the data in Section 2.13. Below is the discussion we included in Section 3.5:

"We further compare MUSICAv0 and WRF-Chem results with surface $PM_{2.5}$, CO, $NO_2$, and $O_3$ measurements from SAAQIS in South Africa (Figures S8-S11). Overall, the performance of MUSICAv0 and WRF-Chem compared to SAAQIS data are similar. Both models underestimate surface CO in most sites (consistent with the comparisons with satellites) with exceptions near Gauteng (industrialized and urbanized region). Compared to SAAQIS sites near Cape Point, MUSICAv0 does not show overestimation which is opposite to the overestimation compared to WDCGG Cape Point site. The maximum value of monthly CO observations from WDCGG Cape Point site in 2017 is ~150 ppb whereas the seasonal mean values of SAAQIS CO measurements near Cape Point site can be up to 600 ppb. SAAQIS CO measurements near Cape Point shows

relatively large spatial variability, , indicating (1) that there may be a wide range of emission sources that are poorly captured by the model and (2) a large role of local sources and potentially complex meteorology. In addition, uncertainties in observations could also contribute to the difference. Both models tend to overestimate $NO_2$ near Gauteng, which may be related to local emissions. Both models can either overestimate or underestimate $PM_{2.5}$ and/or $O_3$ at different SAAQIS sites. The model bias in $PM_{2.5}$ and $O_3$ shows large spatial variability especially near Gauteng. Higher model resolution is needed to address the highly complex and diverse environment in the region. Lastly, it is worth pointing out that in South Africa, both models have evident bias in $PM_{2.5}$ near Gauteng (Figure S11) however modeled AOD from both models agree relatively well with MODIS and AERONET (Figure 4). More studies are needed to understand this feature."

12. Section 3.2: There are DACCIWA measurements that cover at least part of 2017. These could also be used in the model validation.
AOD: https://acp.copernicus.org/articles/21/1815/2021/acp-21-1815-2021.pdf
https://acp.copernicus.org/articles/18/6275/2018/
**Response**: We included surface PM2.5 measurements in West Africa and AERONET AOD in the study.
We added description of the PM2.5 data and AERONET in Sections 2.7 and 2.13, respectively. Below is the discussion we included in Section 3.2:
"We compare the models with weekly $PM_{2.5}$ measurements at 3 sites in Abidjan (representing domestic fires emissions, waste burning at landfill, and traffic) and 1 site in Cotonou representing traffic emissions (Figure S7). Overall, both models underestimate $PM_{2.5}$ at the three Abidjan sites, especially near the domestic fire emissions where measured $PM_{2.5}$ exceeded 400 μg/m$^3$. We include open burning emissions in the MUSICAv0 simulation however the significant underestimation point to the possibility of missing emissions. Moreover, these three sites in Abidjan are within the same city and near strong emission sources and hence are challenging for both models to resolve. In fact, they fall into the same model grids and therefore model values at the three sites are the same for both models. This demonstrates the need of higher model resolution to resolve variabilities of air quality in a city."

13. Lines 310-311: "Similar to North Africa, the model biases in AOD over West Africa are also likely driven by dust and biomass burning." Only dust was noted in N Africa. Are there no dust emissions in the emission inventory? Reading later on it does seem dust is included. As noted in methods, I recommend adding a little more detail about the dust modeling as it is an important point.
**Response**: Dust emissions in MUSICAv0 and WRF-Chem are not included in emission inventories. They are calculated online in the model. We added description of dust schemes in MUSICAv0 and WRF-Chem in the manuscript. Please see response to Reviewer 2 Comment #8 for details.

14. Line 329: Figure S7.
Journal to confirm, but I would expect Supp Material to be in the order that it is refered to in the text. I can see Figure S1 (for example) is refered to below for the first time. I believe the reason they are not in order in the text is because they are in a similar order to the figures in main text

(perhaps?). So that is why I am saying the journal must clarify if there is a specific convention to follow.

**Response**: Figure S1 is referred in Section 3 for the first time before other figures in the supplement ("Spatial distributions of model biases against the OMI tropospheric NO2 column, MODIS AOD, and OMI tropospheric HCHO column are included in Figures S1–S3").

15. Lines 343-345: "Indeed, the emission estimates from the newest version of FINN (FINNv2.5; Wiedinmyer et al., 2023) are higher compared to both QFED (used in the MUSICAv0 simulation) and FINNv1.5 (used in the WRF-Chem simulation) in this region." I am not seeing where this is shown in the figures. Or is it from another paper?

**Response:** The reference for FINNv2.5 has been updated in the revised manuscript:
Wiedinmyer, C., Kimura, Y., McDonald-Buller, E. C., Emmons, L. K., Buchholz, R. R., Tang, W., Seto, K., Joseph, M. B., Barsanti, K. C., Carlton, A. G., and Yokelson, R.: The Fire Inventory from NCAR version 2.5: an updated global fire emissions model for climate and chemistry applications, EGUsphere [preprint], https://doi.org/10.5194/egusphere-2023-124, 2023.

Figure 4 in Wiedinmyer et al. (2023) (also attached below) shows that CO and NOx emissions from FINNv2.5 are higher than FINNv1.5 and QFED.

[Figure]

16. Lines 348: "Figure S4" this is not the correct. I think it should be Figure S3. ..

**Response**: Thank you. We changed "Figure S4" to "Figure S3".

17. Line 349: "and along the west coast of Southern Africa". This is harder to see. Is this referring to Namibia?

**Response**: We deleted "and along the west coast of Southern Africa".

18. "co-locates" in time and space as the seasonal cycle follows the fire seasonal cycle.

**Response**: We changed "underestimation of HCHO in Central Africa (Figure S4) co-locates with the underestimation of CO during fire season (Figure S1), implying that fire emissions may also

contribute to the HCHO underestimation in MUSICAv0" to "the underestimation of HCHO in Central Africa (Figure S4) co-locates with the underestimation of CO in time and space (Figure S1), implying that fire emissions that contributed to model CO biases may also contribute to the HCHO underestimation in MUSICAv0 during fire season".

19. Line 382-384: "The Nairobi site is an exception where both MUSICAv0 and WRF-Chem simulations significantly overestimate O3 in all seasons (mean bias of MUSICAv0 and WRF-Chem below 200 hPa are 27 ppb and 20 ppb, respectively)." Are the IAGOS measurements also in Nairobi? It is hard to tell if any end on Nairobi or go past it in Figure 1. If there is, then it could be interesting to understand how the IAGOS and ozonesonde compare - this is in contrast to South Africa where the IAGOS and ozonesondes are in very different places.

**Response**: Thank you. We looked at IAGOS over Nairobi and found there are only data above 300 hPa in December-January-February (see below). The limited data shows that both MUSICAv0 and WRF-Chem simulations significantly overestimate O3 above 300 hPa in December-January-February, which is consistent with the results of model comparisons with ozonesonde.

[Figure]

20. Lines 398-399: "In addition, model resolutions could also be a potential reason for the underestimation."

As these sites are both in urban areas, I think this point is important and could be expanded with a sentence or two. PM concentrations can vary greatly within an urban environment. Even though the measurements are not from 2017, the data from the AirQo devices could give some indication of this variability to provide support for this assessment.

**Response**: We added the following statement and references to Section 3.4:

"Over Kampala, high spatial variability of $PM_{2.5}$ over the urban environment can contribute to model bias (Atuhaire et al., 2022), as also shown by the AirQo low-cost air quality monitors (Sserunjogi et al., 2022; Okure et al., 2022)."

Atuhaire, C., Gidudu, A., Bainomugisha, E. and Mazimwe, A., 2022. Determination of Satellite-Derived PM2. 5 for Kampala District, Uganda. Geomatics, 2(1), pp.125-143.

Okure, D., Ssematimba, J., Sserunjogi, R., Gracia, N.L., Soppelsa, M.E. and Bainomugisha, E., 2022. Characterization of ambient air quality in selected urban areas in uganda using low-cost sensing and measurement technologies. Environmental Science & Technology, 56(6), pp.3324-3339.

Sserunjogi, R., Ssematimba, J., Okure, D., Ogenrwot, D., Adong, P., Muyama, L., Nsimbe, N., Bbaale, M. and Bainomugisha, E., 2022. Seeing the air in detail: Hyperlocal air quality dataset collected from spatially distributed AirQo network. Data in Brief, 44, p.108512.

21. Lines 402-405: "Among the five regions, MUSICAv0 has the lowest mean bias (-3.2´1017 molecules/cm2 annually) over Southern Africa (Figure 3). WRF-Chem also has low mean bias and RMSE over Southern Africa except for the months of September, October, and November (SON) period where WRF-Chem has larger CO mean bias (-6.2´1017 molecules/cm2) than MUSICAv0."
in CO, correct?
**Response**: Thank you. It is CO and we have revised the statement.

22. Line 412-414: "CO tracers in the model (Figures S4 and S5) show that CO at Cape Point is mainly driven by anthropogenic CO emissions from Southern Africa."
Papers on the characterization of this site should be read to see that this site is influenced by marine and anthropogenic air masses (e.g. https://www.tandfonline.com/doi/abs/10.1080/0035919X.2018.1477854?journalCode=ttrs20). I am not questioning what the tracer says, but rather that this conclusion is not necessarily a true reflection of the impacts at this site. It could help to highlight here that Cape Point is not a really representative site for southern Africa, as it is on the far tip of southern Africa and has such strong impact from clean marine. It is a GAW site, so it is understood why it was included, but still, it does have limitations and these should be noted. This is another reason why it is puzzling why other measurements in South Africa were not also included when they are available. This is a gap in the manuscript that I strongly recommend is addressed.
**Response**:
To address the reviewer's comment on the representativeness of Cape Point, we have modified the discussion in Section 3.5 as suggested by the reviewer:
"CO tracers in the model (Figures S3 and S4) show that the simulated CO at Cape Point is mainly driven by anthropogenic CO emissions from Southern Africa. Therefore, the overestimation of CO at Cape Point by MUSICAv0 may be due to an overestimation of emissions in South Africa. Note that the Cape Point measurement site is located on the tip of southern Africa and has a strong impact from clean marine air (Labuschagne et al., 2018), which the model likely cannot represent accurately."

23. "The Irene ozonesonde site is located in Southern Africa (Figure 1b). Compared to the ozonesonde O3 profiles at the Irene site, however, the MUSICAv0 performance has a seasonal variation (Figure 9e-9h)."
It is hard to tell if the differences across the season are the same or different between IAGIOS (which seems to have been just Cape Town) and Irene looking at the figures. It would help if this could be more explicitly stated or quantified in the text. These are very different sites, so it would not be surprising to see differences. The ozone at the Irene site has been very well characterized and explained going back to the SAFARI campaigns.
**Response**: Thank you for pointing this out. We have revised the paragraph:
"Over Southern Africa, MUSICAv0 tends to overestimate $O_3$ compared to IAGOS at all levels at all seasons in 2017 (Figure 7). The MUSICAv0 $O_3$ bias is 5-10 ppb below 800 hPa for the four seasons and 23-39 ppb at 225 hPa. The concentration of O3S over Southern Africa is higher than those over other regions. However, the correlation of $O_3$S and model $O_3$ bias is lower than other regions (0.13) indicating stratosphere-to-troposphere flux of ozone may not be the main driver of $O_3$ bias over Southern Africa even though stratosphere-to-troposphere flux of ozone is relatively

strong in the region (e.g., Leclair De Bellevue et al., 2006; Clain et al., 2009; Mkololo et al., 2020). The Irene ozonesonde site is located in Southern Africa (Figure 1b). Compared to the ozonesonde $O_3$ profiles at the Irene site, however, the sign of MUSICAv0 has a seasonal variation (Figure 9e-9h). For example, at 675–725 hPa, MUSICAv0 $O_3$ bias in MAM and JJA is 3-9 ppb whereas in SON and DJF it is -2 to -6 ppb. The IAGOS measurements and the Irene ozonesonde site are not co-located, so the difference is expected due to the different sampling locations and environment. Compared to other ozonesonde sites, the correlation of $O_3S$ and model $O_3$ bias over Southern Africa is lower (0.14) and MUSICAv0 agrees relatively well with observations, which is consistent with the comparison results with IAGOS data (Figure 7)."

We also added the following statement to IAGOS data section:
"The IAGOS instruments are onboard commercial airplanes and the sampling may not be representative of the whole sub-regions. For example, IAGOS data over southern Africa only covers the west part of southern Africa."

24. Lines 456-457: "The changes in CO bias over the Southern Hemisphere are likely due to seasonal change in fire emissions."
With a larger bias during the burning season it seems. From this a points above where there are issues in simulated during the burning season, it seems that this very key source is still not well-represented. I would recommend adding this to the discussions and conclusion - as for Africa, this is a key source. As noted above, dust, too.
**Response**: We added the following statement to the Conclusion:
"Fire and dust are important sources of air pollution in Africa. The performance of MUSICAv0 is degraded during fire season and over dust regions. Uncertainties in emission estimates of fire and dust and in the model representation of atmospheric processes could potentially contribute to the model biases. Future studies on fire and dust in Africa are needed to address these uncertainties and air quality modeling over Africa."

25. Lines 466-468: "The positive bias in O3 decreases from the upper troposphere towards the surface, indicating that the overestimation of O3 in the troposphere may be due to stratosphere-to-troposphere flux of ozone."
This is just a suggestion - perhaps this paper could be helpful as it shows the link in transport of ozone across SH? https://acp.copernicus.org/articles/14/9855/2014/acp-14-9855-2014.pdf
**Response**: We added the following statement to Section 3.6:
"Thompson et al. (2014) found $O_3$ at the Irene site is also influenced by long-range transport of growing pollution in the Southern Hemisphere, which could also contribute to the model bias."

26. Section 4: This is a really interesting approach. It does seem that a lot of Africa has higher values. Is this larger than other areas? Has something like this been applied to a region that is better characterized by models (e.g. US or Europe)? I understand that the point is to highlight where in Africa is the value higher, but I think it could be really helpful to also just provide a comparison (if possible) to US or EU to see who well or poorly the models perform in Africa. I am not suggesting a further analysis, but rather is this information available? Then it could help to be able to put these results from African in context before going to this region in E Africa.
**Response**: Thank you. Taylor score is widely used to quantify model skills. However, the Taylor-score-based approach in this study is developed here and has not been applied to other regions.

Therefore, such information is not available and we are not able to provide more information on US or EU at this point for this study. Nevertheless, we agree with the reviewer that such information could help put these results from African in context with a global reference. In the future study, when we evaluate MUSICAv0 for the global domain, we will try to apply the same analyses to Africa in comparison with other continents if applicable.

27. Lines 499-501: "*Overall, both MUSICAv0 and WRF-Chem have low total Taylor scores in the 30°E – 45°E, 5°S – 5°N region in East Africa (a region of 15° longitude ´ 10° latitude) during MAM (March,*
*April, and May), JJA (June, July, and August), and SON (September, October, and November),*
*…"*
Was the average compared across seasons? Or annually? Just looking at the figure, I can see that this area has low values, but it doesn't automatically stick out as the worst as there is a lot of areas with the darker color. In the WRF-Chem annual, for example, southern Africa looks darker. So I recommend just providing some more specific motivation for why this area was selected as having a particularly low Taylor score. I do see population and vegetation is also highlighted as reasons, but there are other areas in the domain that this is also true.
**Response**: Seasonal maps are compared because we would like to provide information on both location and season of potential field campaign(s). During DJF, Taylor score is the lowest over Southern Africa. However, in other seasons as described in the manuscript, Taylor score over the 30°E – 45°E, 5°S – 5°N region is among the lowest. We added the following statement to avoid confusion:
"Besides the 30°E – 45°E, 5°S – 5°N region highlighted in Figure 13, there are a few other regions with low Taylor scores for both MUSICAv0 and WRF-Chem such as 10°E – 20°E, -30°S – -20°N region and the east of Madagascar."

28. Section 5 Conclusions: I strongly recommend that limitations are discussed in the conclusion in order to put these results in context. I understand why a regional analysis approach was taken as this is a very large domain. However, this does mean that highly complex and diverse areas are lumped together and discussed together. Treating such large and diverse areas as homogeneous in the model - measurement comparisons does lose key details. I am not recommending that more highly resolved analyses are needed, but rather that this point is discussed. I strongly recommend that the limitations of doing this and the impacts on the results should be acknowledged and discussed here.
**Response**: We agree with the reviewer that Africa is a very large domain with highly complex and diverse areas. Each area may have different dominant factors of air quality and different scientific questions that need to be addressed. We divided the continent to five sub-regions (North Africa, West Africa, East Africa, Central Africa, and Southern Africa) to address the diversity in atmospheric chemistry environment to some degree. The primary goal of this paper is to evaluate the model over Africa and show the overall performance of MUSICAv0 over sub-regions of Africa rather than addressing specific scientific questions. We understand each sub-region is not homogeneous. In fact, different cities in the same sub-region may have different emission characteristics. In the future we will look into more details with smaller domains when specific scientific questions are studied. To address the reviewer's comment on inhomogeneity, we added the following discussion to the conclusion:

"We divided the continent into five sub-regions to show the overall performance of MUSICAv0 over sub-regions of Africa. This accounted for the diversity in atmospheric chemistry environment to some degree. However, each sub-region is not homogeneous. In fact, different cities in the same sub-region may have different emission characteristics. In the future when specific scientific questions are studied with MUSICAv0, we will use higher resolution to address the highly complex and diverse environment."

29. Line 569: "missing local sources" Also as noted above, also can be due to lack of data as one site in a city is not representative of the full city.
**Response**: Thank you for pointing this out. We added the following statement to the conclusion:
 "In addition, lack of data could also contribute to disagreement in model and in situ observations as one site in a city is not representative of the full city".

30. Line 570: "Field campaigns and more in situ observations" This is true across Africa isn't it?
**Response**: We change the statement "30°E–45°E, 5°S–5°N region in East Africa" to "Field campaigns and more in situ observations in 30°E–45°E, 5°S–5°N region in East Africa (as well as other regions in Africa)".

31. "In the future, we plan to conduct a model simulation for multiple years and develop additional model grids with potentially higher resolution in Africa sub-regions based on the current MUSICAv0 Africa grid."
As noted in other places, a gap in this study is not using many of the measurements that are available for model validation. Would further studies then also look to include more measurements from the region? Moving to later years cold help as the monitoring is increasing. Also, higher resolution and smaller domains in the modelling could make use of existing networks easier (as many are located close together).
**Response**: In our future model analyses, we will run the model for more recent years and compare with more available data. We added the following statement to the conclusion:
"Higher resolution will benefit the comparisons of model and in situ observations. The future simulation will be conducted for years after 2017 as there are more in situ observations available in recent years."

32. Supplement: Figure S6 caption: These have been estimated using local data for South Africa. A comparison could help as another point of comparison. It is not for 2017, but is a multi-year average. http://www.scielo.org.za/pdf/sajs/v117n9-10/22.pdf
**Response**: Thank you. We compared our model results with the multi-year average (2008-2015) from Maseko et al. (2021). The figures are attached below. We added the statement below to Section 3.2:
"We also compared our modeled lightning NO emissions with a multi-year average climatology (2008-2015) from Maseko et al. (2021) over South Africa, and found that the seasonal cycle from MUSICAv0 and standard CAM-chem are consistent with the climatology. The magnitude of MUSICAv0 lightning NO emissions overall agree better with the climatology compared to that from standard CAM-chem simulation."

Figure 4 from Maseko et al. (2021):

[Figure]

Simulations from this study:

[Figure]

33. Supplement: Figure S6 caption: what are red circles? I think ozonesondes?
**Response**: Yes, it is ozonesonde. We added the following statement to caption of Figure S6:
"Four ozonesonde sites are shown by red circles (Ascension, Irene, Nairobi, and La Reunion)."